# Loss of E-cadherin is causal to pathologic changes in chronic lung disease

Baishakhi Ghosh[1], Jeffrey Loube[1], Shreeti Thapa[2], Hurley Ryan[1], Erin Capodanno[3], Daniel Chen [3], Carter Swaby[3], Si Chen[2,5], Saborny Mahmud[2], Mirit Girgis [3], Kristine Nishida[2], Linyan Ying[2,6], Pratulya Pragadaraju Chengala[1], Ethan Tieng[3], Michael Burnim[2], Ara Wally[2], Debarshi Bhowmik[2], Michael Zaykaner[4], Bonnie Yeung-Luk[1], Wayne Mitzner [1], Shyam Biswal[1] & Venkataramana K. Sidhaye [1,2✉]

Epithelial cells line the lung mucosal surface and are the first line of defense against toxic exposures to environmental insults, and their integrity is critical to lung health. An early finding in the lung epithelium of patients with chronic obstructive pulmonary disease (COPD) is the loss of a key component of the adherens junction protein called E-cadherin. The cause of this decrease is not known and could be due to luminal insults or structural changes in the small airways. Irrespective, it is unknown whether the loss of E-cadherin is a marker or a driver of disease. Here we report that loss of E-cadherin is causal to the development of chronic lung disease. Using cell-type-specific promoters, we find that knockout of E-cadherin in alveolar epithelial type II but not type 1 cells in adult mouse models results in airspace enlargement. Furthermore, the knockout of E-cadherin in airway ciliated cells, but not club cells, increase airway hyperreactivity. We demonstrate that strategies to upregulate E-cadherin rescue monolayer integrity and serve as a potential therapeutic target.

[1] Department of Environmental Health and Engineering, Johns Hopkins Bloomberg School of Public Health, Baltimore, MD, USA. [2] Department of Pulmonary and Critical Care Medicine, Johns Hopkins School of Medicine, Baltimore, MD, USA. [3] Johns Hopkins University, Baltimore, MD, USA. [4] Department of Otolaryngology-Head and Neck Surgery, Johns Hopkins School of Medicine, Baltimore, MD, USA. [5] Present address: Department of Pulmonary and Critical Care Medicine, Shanghai East Hospital, Tongji University, Shanghai 200120, China. [6] Present address: Department of Respiration, Children's Hospital of Chongqing Medical University, Chongqing, China. ✉email: vsidhay1@jhmi.edu

Maintaining the protective lung mucosal surface in the adult lung is a critical first line of defense as it encounters over 10,000 L of air daily with over 100 billion particles, chemicals, and pathogens[1]. The healthy epithelium is a robust barrier maintained through the physical linkage of its constituent cells to form a mechanically integrated, multicellular structure. This linkage is established by a series of cell-cell contacts, of which E-cadherin plays a central role, with crucial roles in tissue morphogenesis and post-developmental tissue homeostasis[2,3]. E-cadherin is an adherens junctional protein with well-appreciated functional cooperativity with the actin cytoskeleton[4–6]. It is involved in the regulation of the architecture of the epithelia, regulating the paracellular permeability, and suppressing intracellular signaling[7,8].

There is increasing evidence that disruptions in the physical structure of the epithelium are critical early changes in the development of many chronic lung diseases, including chronic obstructive pulmonary disease (COPD)[9,10] with transcriptomics analysis indicating a "loss of epithelial coherence," decline of regenerative repair as the initial defects in disease pathogenesis, and early changes in expression of genes involved in the actin cytoskeleton and cell adhesion being spurring[10]. We and others have previously reported that chronic cigarette smoke (CS) exposure to primary human bronchial epithelial cells in an air-liquid interface reduces E-cadherin, and patients with COPD have less E-cadherin[4,11–15]. In addition, the decrease of E-cadherin is associated with the development of COPD[16–20] with COPD tissues displaying loss of E-cadherin associated with dedifferentiation of epithelium with evidence of subepithelial fibrosis[18]. Moreover, there is regional loss of E-cadherin in emphysematous regions in human lung histologic specimens of COPD, suggesting decreased or degraded E-cadherin, indicating its role in maintaining the integrity of pulmonary epithelium[19,20]. Finally, loss of lung epithelial E-cadherin in utero can impact lung development[21]. However, despite this circumstantial evidence, it is not known if loss of E-cadherin in the adult, fully differentiated lung is truly causal to the tissue remodeling and lung pathology seen in chronic lung diseases such as COPD.

In this study, we investigated if loss of E-cadherin is causal to disease using a combination of genetically manipulated mouse models and primary mouse and human cell in vitro models. We implemented a cell type-specific promoter using Cre/LoxP mice system to knock out E-cadherin in ciliated and alveolar epithelial cell (Type 1 and Type 2) populations in adult mouse models to further investigate its causative role in lung injury by quantifying lung function and lung morphometry. In addition, we assessed if loss of E-cadherin can cause epithelial dysfunction in primary non-diseased human bronchial epithelium (normal) like that seen in human bronchial epithelial cells derived from patients with COPD (COPD cells) to improve our mechanistic understanding of its role in causing lung dysfunction. We further investigated the strategies to upregulate *CDH1* (encodes for E-cadherin) in COPD cells, and CS-injured normal cells can rescue epithelium dysfunction. In this report, we provide evidence that E-cadherin plays a causal role in the pathogenesis of emphysema and airway dysfunction in patients with COPD.

## Results

**In vivo E-cadherin knockdown in mice lung increases lung morphometry and decreases lung function**. To determine if E-cadherin has a causal role in airway dysfunction and parenchymal modeling, we have performed bimonthly intratracheal instillations of adenoviruses Ctrl or Cre recombinases on *Cdh1^{fl/fl}* mice and lung morphometry, and lung function was assessed. Exposure to adeno-Cre causes a reduction in E-cadherin in

mouse lungs, as confirmed by immunofluorescence after 10 days (Fig. 1a), which persisted throughout the instillations period (Fig. 1b, c). After 1-month instillation, the hemotoxylin and eosin (H&E) staining showed increased terminal airway/airspace enlargement (Fig. 1d) with higher mean linear intercept (MLI, Lm) in the Cre delivery group (Fig. 1e), indicating loss of E-cadherin in lung epithelium induces emphysematous injuries. Furthermore, the adeno-Cre group had significantly higher total lung capacity (Fig. 1f) without any difference in the residual volume (Fig. 1g). Lung compliance, which measures the ability of a lung to stretch and expand, was significantly higher in the adeno-Cre group and is consistent with airspace enlargement (Fig. 1h).

We performed a time course to characterize the emphysema that occurred with E-cadherin knockdown. We observed a reduction in E-cadherin in the adeno-Cre instilled group, as confirmed by immunofluorescence (Fig. 1i, j). The H&E staining showed patchy regions. It increased terminal airway/airspace enlargement with increased MLI as observed in the adeno-Cre delivery group, like in the 1-month instilled mice group (Fig. 1k, l). In addition, we observed significantly higher total lung capacity and residual volume, with higher lung compliance in 2-month and 3-month instillations (Fig. 1m–o). These data suggest injury with emphysema in the mice's lungs.

**E-cadherin knockdown in the alveolar type II, but not type I, mouse epithelial cells cause emphysema**. Since E-cadherin regulates the structure and function of the lung epithelium, we further assessed if loss of E-cadherin in specific cell types, particularly in alveolar type I (AT1) and type II (AT2) cells, can contribute to the injuries as observed in the whole lung model. To answer this, we have generated tamoxifen-inducible specific mice to knock out E-cadherin in AT1 cells, using the Ager promoter (*Cdh1^{fl/fl}-Ager^{Cre}*) and AT2 cells, using the Spc promoter (*Cdh1^{fl/fl}-Spc^{Cre}*).

As the AT1 cells cover 95% of the alveolar cells[22], we evaluated the consequences of knocking down E-cadherin in these cells. To study the effects of E-cadherin knockdown in AT1 cells, *Cdh1^{fl/fl}-Ager^{Cre}* received either tamoxifen (TAM) or a normal diet (ND), with a comparison control group of *Cdh1^{fl/fl}-Ager_{WT}* receiving TAM. The total lung capacity, compliance, and residual volume were significantly reduced in *Cdh1^{fl/fl}-Ager^{Cre}* receiving TAM compared to the control group (Fig. 2a–c). Notably, we did not observe any difference in MLI based on lung histology analysis (Fig. 2d, e), indicating E-cadherin knockdown in AT1 cells has a minimal role in lung injury as observed in whole lung knockdown. Interestingly, we observed an increase in mRNA expression of mesenchymal (*Cdh1, Cdh2, Slug, Snai1,* and *Twist1*) and collagen (*Col3a1*) markers without an overall change in E-cadherin expression (*Cdh1*) (Fig. S1). In addition, the Masson trichrome stain also shows an increase in blue staining around the airways of AT1 cells with knocked down E-cadherin, indicating fibrosis (Fig. S2).

To knock down E-cadherin in AT2 cells, *Cdh1^{fl/fl}-Spc^{Cre}* received a TAM diet compared with *Cdh1^{fl/fl}-Spc^{Cre}* receiving ND as a control. The total lung capacity and compliance were significantly increased in *Cdh1^{fl/fl}-Spc^{Cre}* receiving TAM (Fig. 2f, g), with no differences in residual volume (Fig. 2h). Moreover, the MLI analysis of the lungs stained with H & E indicates destruction in enlarged airspaces, indicating emphysematous injuries (Fig. 2i, j).

**Loss of E-cadherin reduces the regenerative capacity of mouse lungs**. Type II epithelial cells are the progenitor cells in the alveoli and are thought to proliferate and repopulate the Type I epithelia

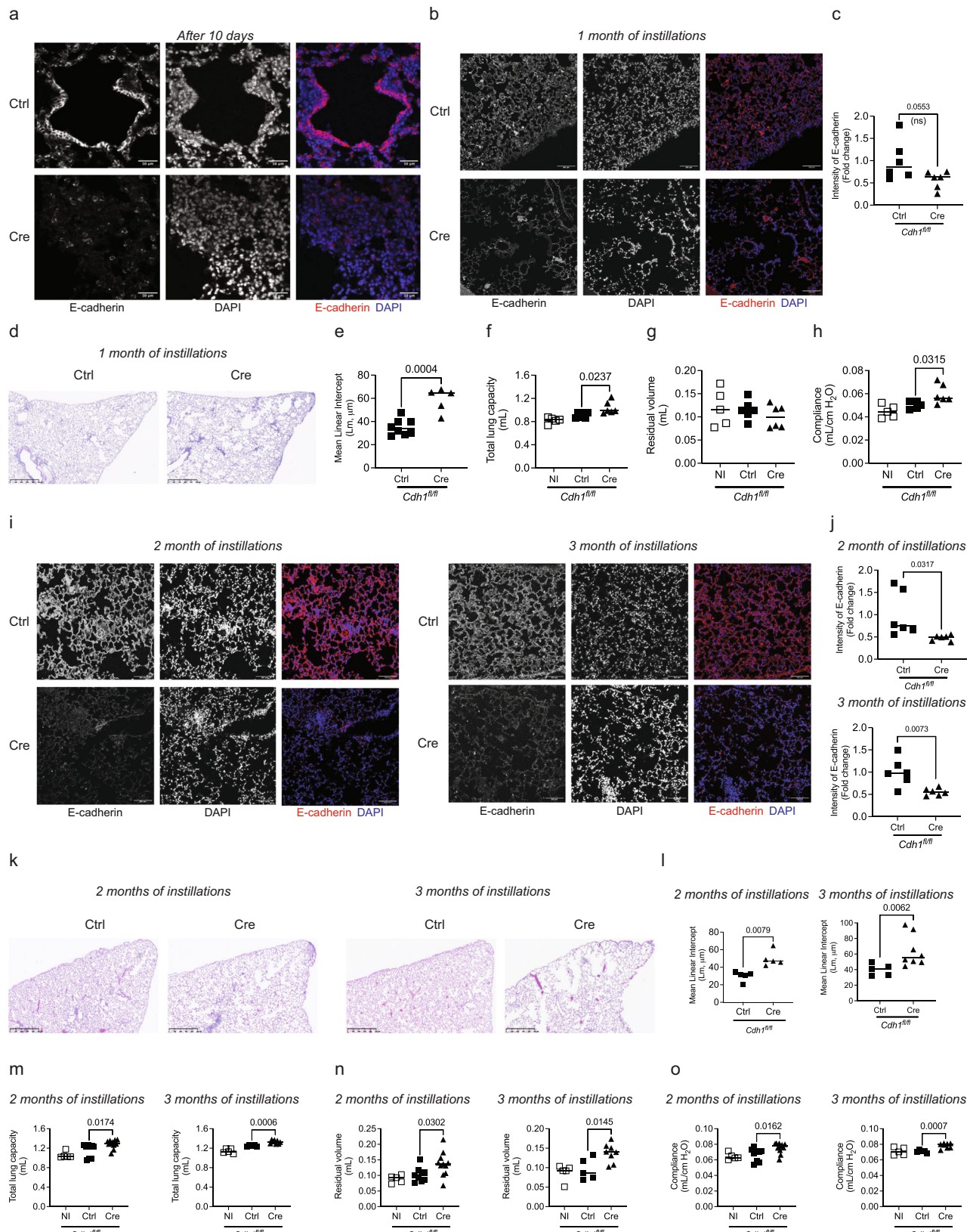

after injury. As we found emphysematous changes specifically after E-cadherin knockdown in Type II alveolar cells, we sought to determine if E-cadherin played a role in cellular proliferation. To characterize the role of E-cadherin on the regenerative capacity of the epithelium, BrdU (Bromodeoxyuridine) staining was performed, as BrdU specifically labels the cells in S-phase (i.e., proliferating cells)[23]. Our in vivo data suggest that knockdown of E-cadherin in mice instilled with adeno-Cre lowers regenerative capacity with reduced BrdU staining (Fig. 3a–c). Also, the knockdown of E-cadherin in AT2 cells contributes to a decrease in the regenerative capacity of the mice lungs (Fig. 3d–f). Furthermore, we observed that both undifferentiated COPD-patient

**Fig. 1 In vivo E-cadherin knockdown in mice lung contributes to increased lung morphometry and decreased lung function.** Representative immunofluorescence of E-cadherin (Red) and DAPI (Blue) in mice lungs showing knockdown of E-cadherin in adeno-Cre (Ad5CMVCre-eGFP) intratracheally instilled group compared to adeno-Ctrl (Ad-5CMVeGFP) after **a** 10 days (scale bar of 50 μm) and **b** 1 month (scale bar of 100 μm) with 10× objective. **c** Decreased fluorescence intensity of E-cadherin in the mice lungs intratracheally instilled with adeno-Cre compared to adeno-Ctrl. Increased terminal airway/airspace enlargement as observed by **d** H & E staining (representative image) of mice lung parenchyma at 5× (scale bar of 500 μm), and **e** increased mean linear intercepts (Lm). Lung function was analyzed, where knockdown of E-cadherin with 1-month instillations of adeno-Cre compared to adeno-Ctrl in Cdh1 flox mice (*Cdh1^{fl/fl}*) caused **f** total lung capacity was increased, **g** residual volume was not altered, and **h** compliance was increased. Immunofluorescence of E-cadherin (Red) and DAPI (Blue) in mice lungs showing knockdown of E-cadherin among adeno-Cre intratracheal instilled group compared to adeno-Ctrl in *Cdh1^{fl/fl}* mice after 2-months and 3-months at 10× (scale bar of 100 μm) as observed in **i** representative immunofluorescence images (Left panel—2 months of instillations, Right panel—3 months of instillations) and **j** fluorescence intensity of E-cadherin (Top panel—2 months of instillations, Bottom panel—3 months of instillations). Increased terminal airway/airspace enlargement as observed in **k** H & E staining at 5× (scale bar of 500 μm) **l** increased Lm in adeno-Cre as compared to adeno-Ctrl instilled *Cdh1^{fl/fl}* mice (Left panel—2 months of instillations, Right panel – 3 months of instillations). Increased **m** total lung capacity, **n** residual volume, and **o** compliance in E-cadherin knockdown due to 2-months (left panel) and 3-months (right panel) instillations of adeno-Cre in *Cdh1^{fl/fl}* mice compared to adeno-Ctrl. Data is expressed as median bars and generated from 5 to 11 mice per group. Kruskal-Walli's test followed by Dunn's multiple comparison test was performed to compare the lung function tests (total lung capacity, residual volume, and compliance), whereas the Mann-Whitney test was performed to assess fluorescence intensity of E-cadherin and Lm. *P* < 0.05 were considered statistically significant. NI: Not instilled *Cdh1^{fl/fl}* mice.

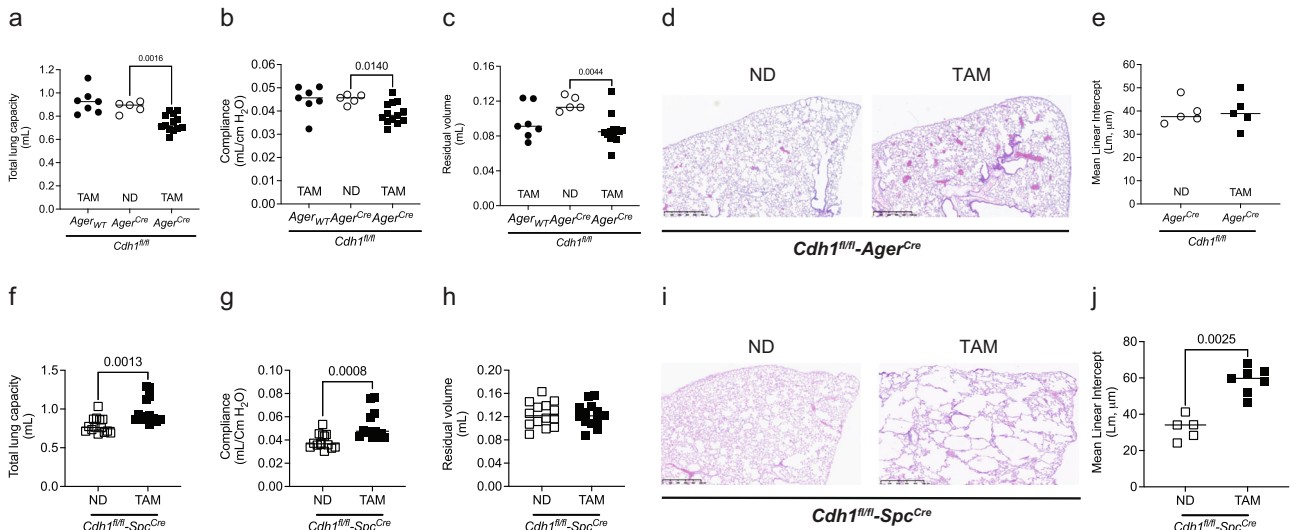

**Fig. 2 In vivo E-cadherin knockdown in the alveolar type, I (AT1) and alveolar type II (AT2) cells have differential effects on mouse lung function and histology.** To knock down E-cadherin in the AT1 cells of mice lungs, *Cdh1^{fl/fl}-Ager^{Cre}* mice were fed a tamoxifen diet (TAM) for 1 month. These were compared to *Cdh1^{fl/fl}-Ager^{Cre}* mice receiving a normal chow diet (ND) and *Cdh1^{fl/fl}-Ager_{WT}* receiving TAM control mice. **a** Total lung capacity, **b** compliance, and **c** residual volume were decreased in E-cadherin knockdown in AT1 cells of mice lung. No difference in lung histology was observed by **d** H & E staining (representative image at 5X with a scale bar of 500 μm) and **e** quantified mean linear intercepts (Lm) after E-cadherin knockdown in AT1 cells. To knock down E-cadherin in the AT2 cells of mice lungs, *Cdh1^{fl/fl}-Spc^{Cre}* mice were fed TAM for 1 month. These were compared to *Cdh1^{fl/fl}-Spc^{Cre}*, mice receiving ND. **f** Total lung capacity was increased, **g** compliance was increased, and **h** without significant changes in residual volume in E-cadherin knockdown in AT2 cells of mice lungs. Lung histology shows airspace enlargement as observed by **i** H & E staining (representative image at 5× with a scale bar of 500 μm) and **j** an increase in Lm after E-cadherin knockdown in AT2 cells. Data is expressed as median bars and generated from 5 to 15 mice per group. Kruskal-Walli's test followed by Dunn's multiple comparison test was performed to assess total lung capacity, compliance, and residual volume in E-cadherin knockdown in AT1 cells. Mann-Whitney test was performed to assess Lm of AT1 and AT2, and lung function tests (total lung capacity, compliance, and residual volume) in E-cadherin knockdown in AT2 cells. Data is generated from 5 to 14 mice per group. *P* < 0.05 were considered statistically significant.

cells and normal cells with E-cadherin (Normal + shCDH1) have fewer BrdU positive cells (Fig. 3g, h), indicating loss of E-cadherin alters the regenerative capacity. Moreover, overexpressing E-cadherin (COPD + CDH1) rescues the proliferative activity with increased BrdU-positive cells (Fig. 3g, h).

**Loss of E-cadherin reduces the regenerative capacity of human airway epithelium.** Although grown under identical conditions, age and sex-matched COPD epithelial cells demonstrate a delay in monolayer formation. The COPD monolayers consisted of fewer total cells and showed a persistent decrease in E-cadherin (Fig. 4). Specifically, we observed fewer ciliated cells (β-tubulin), more goblet cells (MUC5AC), and similar numbers of basal cell numbers (Cytokeratin 14) in COPD as compared to age and sex-matched normal human epithelium (Fig. 4).

**E-cadherin knockdown in ciliated cells, but not club cells, increases airway hyperresponsiveness (AHR).** To knock down E-cadherin in ciliated cells (which express the *FoxJ1* promoter), *Cdh1^{fl/fl}-Foxj1_{Het}^{Cre}* received either a TAM diet or an ND, compared to *Cdh1^{fl/fl}-Foxj1_{WT}^{Cre}* receiving a TAM diet. The lung function parameters (total lung capacity, vital capacity, residual volume, and compliance) and MLI were not affected in *Cdh1^{fl/fl}-Foxj1_{Het}^{Cre}* receiving TAM (Fig. 5a–e). Notably, we observed that *Cdh1^{fl/fl}-Foxj1_{Het}^{Cre}* receiving TAM diet had significantly increased airway hyperreactivity (AHR) than *Cdh1^{fl/fl}-Foxj1_{Het}^{Cre}*

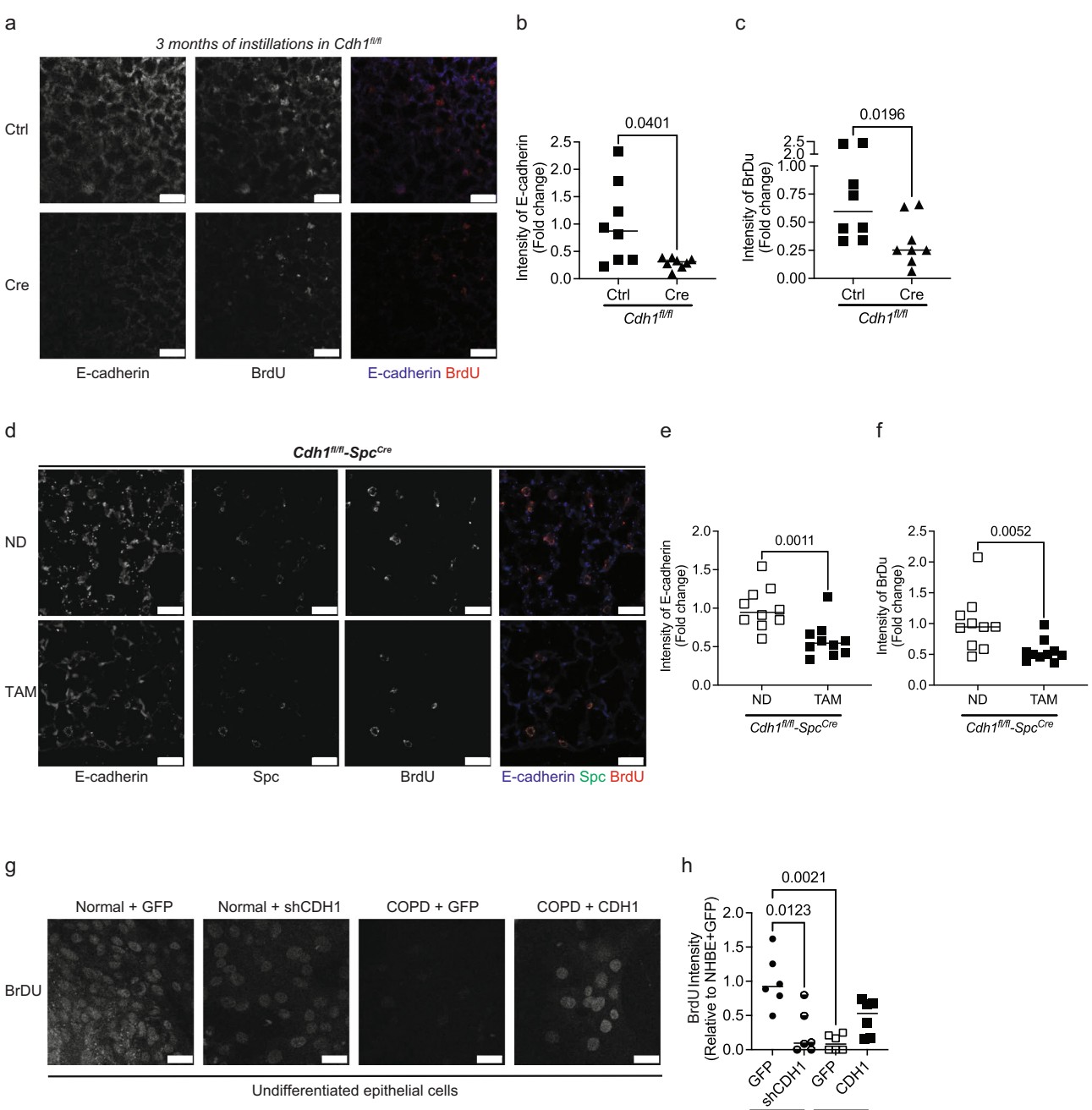

**Fig. 3 Knockdown of E-cadherin decreases the regeneration of epithelium in the mice lungs.** Regeneration of epithelium was assessed by BrdU staining. **(a)** Representative image at 10X (scale bar of 50 μm) of *Cdh1fl/fl* mice instilled with adeno-Cre recombinase (Ad5CMVCre-eGFP) to knockdown E-cadherin for 3 months shows decreases in BrdU as compared to *Cdh1fl/fl* instilled with adeno-Ctrl (Ad-5CMVeGFP). Decreases in the fluorescence intensity of **b** E-cadherin and **c** BrdU in *Cdh1fl/fl* mice instilled with adeno-Cre as compared to *Cdh1fl/fl* mice instilled with adeno-Ctrl. Data is generated from eight mice. To knock down E-cadherin in the AT2 cells of mice lungs, *Cdh1fl/fl-SpcCre* mice were fed tamoxifen (TAM) for 30 days. These were compared to *Cdh1fl/fl-SpcCre* mice receiving a normal chow diet (ND). In vivo knockdown of E-cadherin in *Cdh1fl/fl-SpcCre* mice show decreases in BrdU expression as observed in the **d** representative images at 10X (scale bar of 25 μm), with decreases in the intensity of **e** E-cadherin and **f** BrdU in AT2 cells. Data is generated from five mice. Undifferentiated normal basal epithelial cells were transfected with Ad-GFP-U6-h-CDH1-shRNA to knock down E-cadherin, and undifferentiated COPD cells were transfected with Ad-GFP-U6-h-CDH1 to overexpress E-cadherin. We compared to respective undifferentiated Normal/COPD with Ad-GFP. **g** Representative image at 40× (scale bar of 25 μm) and **h** quantification showing decreased BrdU intensity in the normal epithelium with E-cadherin knockdown (Normal + shCDH1) and COPD at baseline (COPD + GFP) as compared to Normal + GFP. Also, COPD with overexpressed E-cadherin (COPD + CDH1) demonstrates increased BrdU intensity compared to COPD + GFP. Data are expressed as median bars and generated from three inserts from two donors.

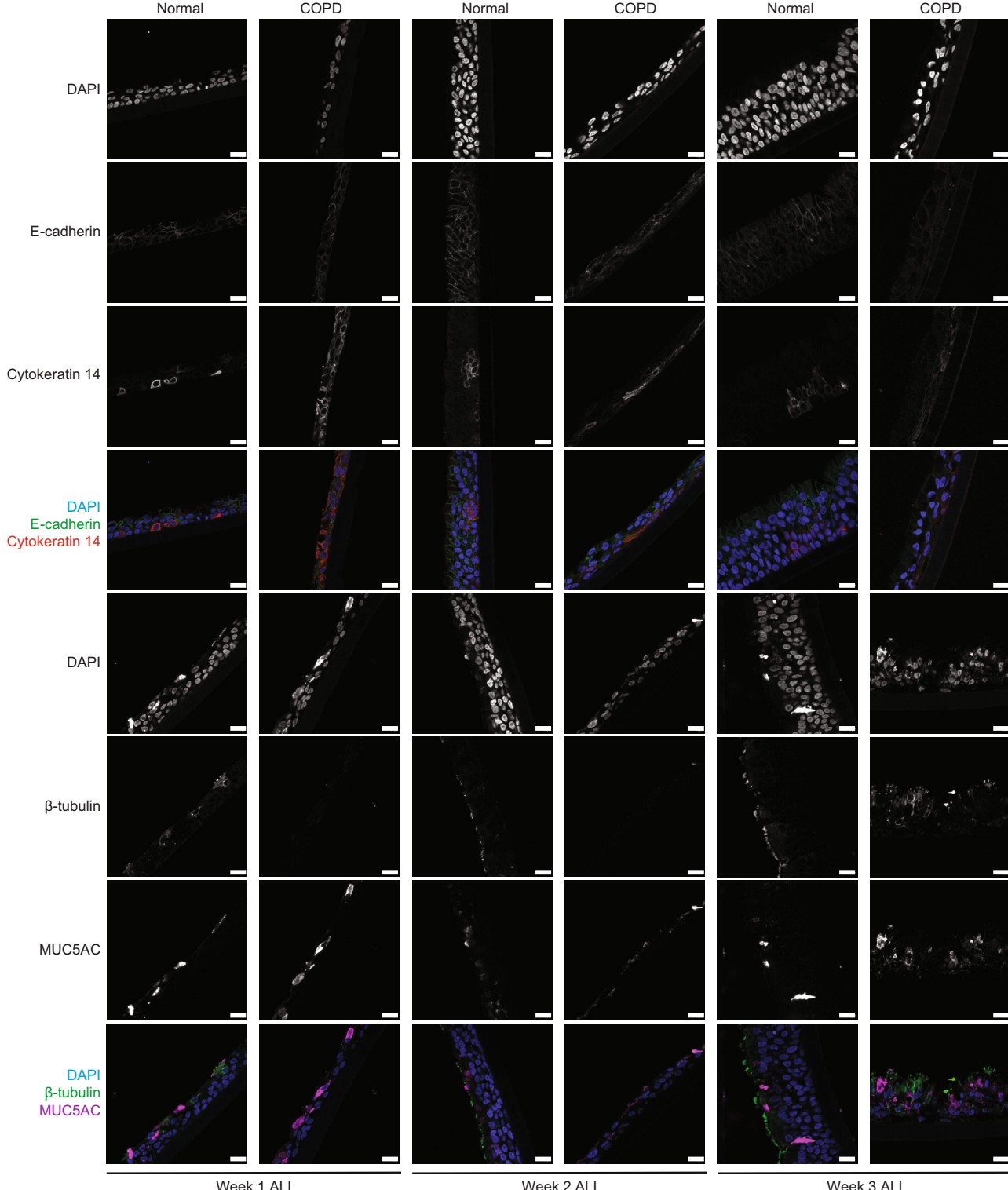

**Fig. 4 Knockdown of E-cadherin decreases epithelial regeneration in human epithelial cells.** Immunofluorescence at 40× (scale bar of 50 μm) of COPD human bronchial epithelial cells differentiated at week one to three of air–liquid interface (ALI) show decreased expression of E-cadherin and β-tubulin (ciliated cells marker) expression, increase expression of MUC5AC (goblet cell marker), without any changes in Cytokeratin 14 (Basal cell marker), as compared to non-diseased human bronchial epithelial (normal) cells.

receiving ND and $Cdh1^{fl/fl}$-$Foxj1_{WT}^{Cre}$ receiving TAM (Fig. 5f), indicating that E-cadherin knockdown in ciliated cells increases AHR.

As club cells can regenerate the injured epithelium and differentiate into other cells of the airway epithelium[24,25], we evaluated the changes in airway reactivity with the loss of E-cadherin (using the Scbg1a1 promoter of club cells). $Cdh1^{fl/fl}$-$Scbg1a1^{Cre}$ received either a TAM diet or ND to knock down E-cadherin in club cells. E-cadherin knockdown in club cells did not alter airway reactivity measurements (Fig. 5g).

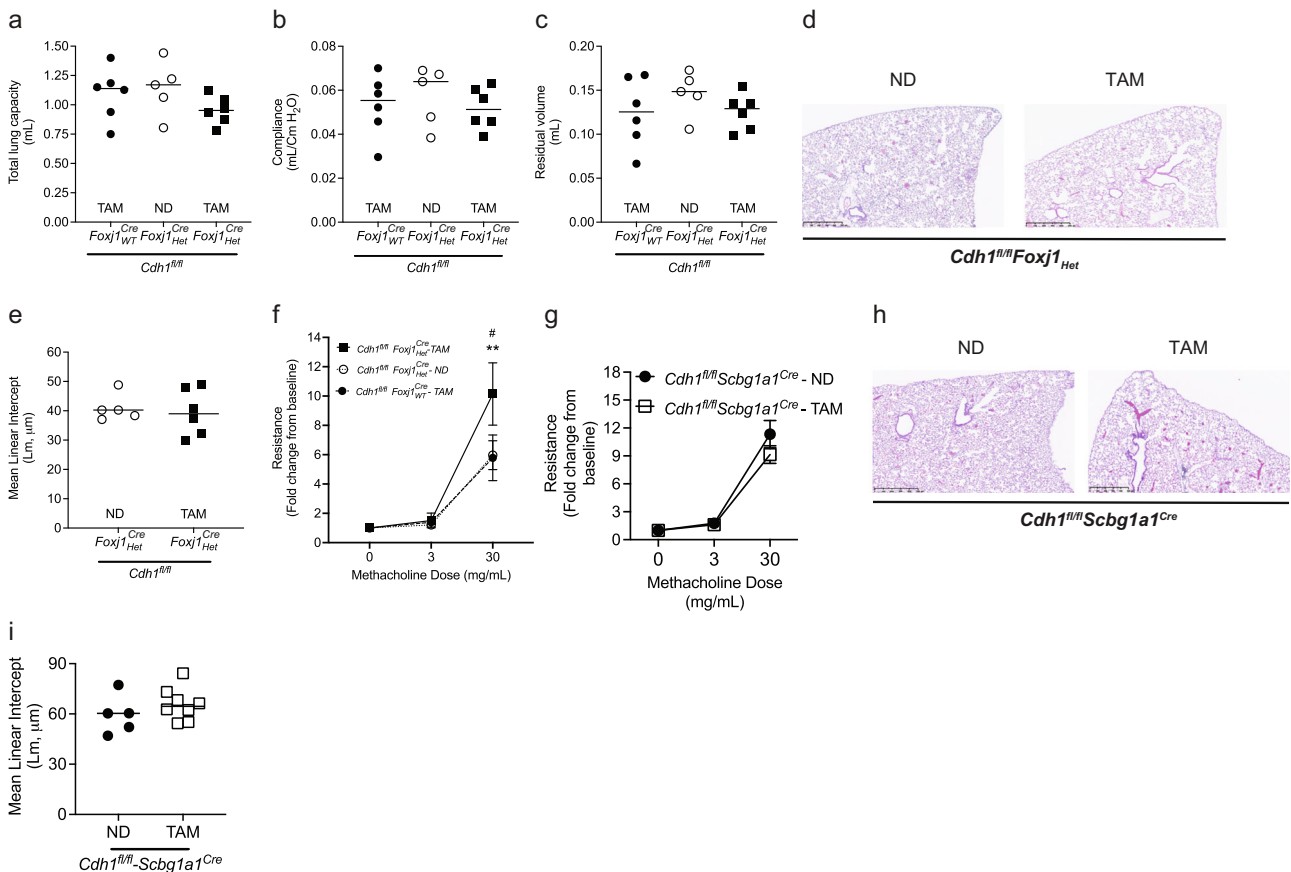

**Fig. 5 E-cadherin knockdown in the ciliated cells, but not club cells, contributes to increased airway hyperresponsiveness.** To knock down E-cadherin in the ciliated cells of mice lungs, $Cdh1^{fl/fl}$-$Foxj1Cre_{Het}$ mice were fed tamoxifen (TAM) for 30 days. These were compared to $Cdh1^{fl/fl}$-$Foxj1Cre_{Het}$ mice receiving a normal chow diet (ND) and $Cdh1^{fl/fl}$-$Foxj1Cre_{WT}$ receiving TAM control mice. No change in **a** total lung capacity, **b** compliance, and **c** residual volume was observed in the ciliated cells with E-cadherin knockdown of mice lungs. Also, no changes were observed in **d** H & E staining (representative image at 5X with a scale bar of 500 μm), and **e** quantified mean linear intercepts (Lm) after E-cadherin knockdown in ciliated cells. **f** Increased airway hyperreactivity (AHR) in ciliated cells with E-cadherin knockdown. Data is expressed as median bars and representative of 5 to 9 mice. To knock down E-cadherin in the club cells of mice lungs, $Cdh1^{fl/fl}$-$Scbg1a1Cre$ mice were fed a TAM for 30 days. These were compared to $Cdh1^{fl/fl}$-$Scbg1a1Cre$ mice receiving an ND. No difference in **g** airway reactivity, and histology as observed in **h** H & E staining (representative image), and **i** MLI among $Cdh1^{fl/fl}$-$Scbg1a1Cre$ receiving a TAM and an ND. Data is expressed as median bars and generated from 5 to 9 mice. Kruskal-Wallis test followed by Dunn's multiple comparison test was performed for total lung capacity, compliance, residual volume, and AHR in E-cadherin knock down in ciliated cells. Mann-Whitney test was performed for airway reactivity in E-cadherin knockdown in club cells and MLI values. $P < 0.05$ were considered statistically significant.

Furthermore, loss of E-cadherin in club cells did not alter the lung histopathology and resulted in no change in MLI (Fig. h & i).

**Loss of E-cadherin reduces epithelial monolayer integrity in mouse and human airway models.** To determine if the increase in AHR observed after knockdown of E-cadherin in ciliated cells was due to altered monolayer function, we cultured mouse tracheal epithelial cells (mTEC) from $Cdh1^{fl/fl}$ mice for 14 days of an air-liquid interface (ALI). We treated these monolayers with either Ad5CMVeGFP (Ctrl) or Ad5CMVCre-eGFP (Cre) adenoviruses, and treatment with the Cre virus caused a ~50% reduction in $Cdh1$ mRNA expression (Fig. 6a) and ~75% reduction in E-cadherin abundance (Fig. 6b). Loss of E-cadherin disrupted the monolayer integrity, with decreased epithelial resistance (Fig. 6c) and increased cellular movement (Fig. 6d). There was no influence on ciliary function as measured by % moving cilia on the monolayer and ciliary beat frequency (CBF) (Fig. S3a, b).

To determine if we saw a similar disruption in monolayer integrity in the human epithelium, we cultured age and

sex-matched normal and COPD. As we have shown previously, the COPD cells have reduced expression of E-cadherin as compared to age and sex-matched normal cells (Fig. S4a)[4,15]. In addition, we confirmed that normal cells exposed to cigarette smoke (CS) demonstrate reduced E-cadherin expression (Fig. S4b)[14,15].

To determine if loss of E-cadherin disrupts monolayer integrity, we knocked down E-cadherin in normal cells using an adenovirus-based shRNA targeting the $CDH1$ (Ad-GFP-U6-shRNA, shCDH1), compared to an adenovirus-based scrambled shRNA with GFP (Ad-GFP-U6-shRNA, GFP). We observed a ~50% E-cadherin knockdown in normal cells by fluorescence (20× magnification), qPCR, and western blot assay (Figs. S5 and 6e, f). This knockdown significantly decreased barrier function (Fig. 5g) with an increasing trend of monolayer permeability (Fig. 5h), contributing to increased cell movement (Fig. 5i). Interestingly, the E-cadherin knockdown did not affect % moving cilia and CBF (Fig. S6). Also, normal patient-derived cells with E-cadherin knockdown showed lower expression of $CDH1$ (encodes for E-cadherin) and higher expression of EMT markers such as $CDH2$, $SNAI2$, $SLUG$, and $TWIST2$ (Fig. S7a–d), suggestive of some cellular plasticity.

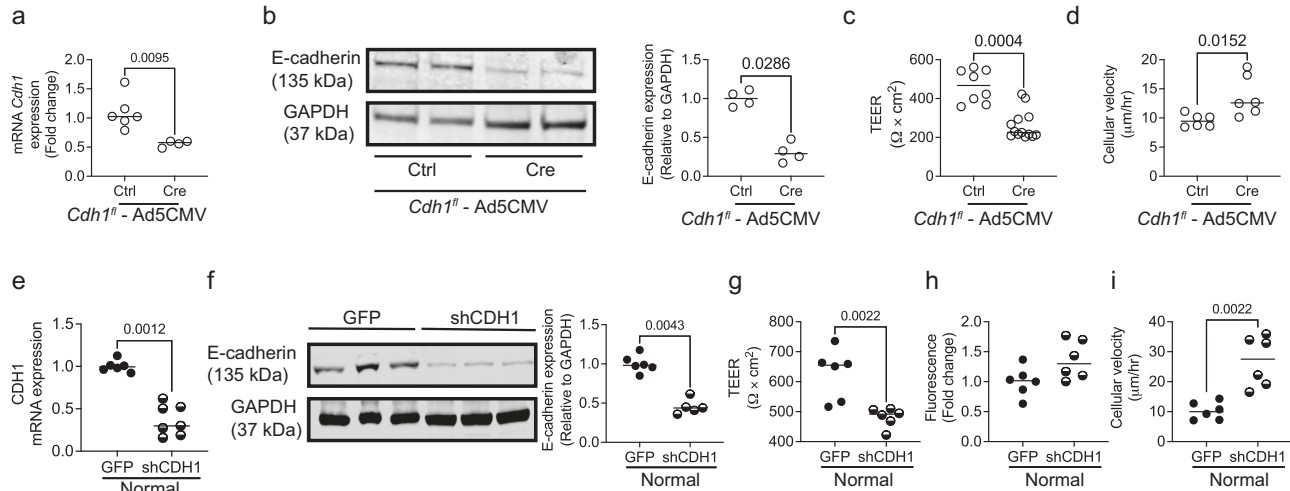

**Fig. 6 Knockdown of E-cadherin contributes to epithelial dysfunction.** To knock down E-cadherin in airways, mice tracheal epithelial cells (mTECs) from *Cdh1fl/fl* mice cultured at the air-liquid interface (ALI) were transfected with Ad5CMVCre-eGFP (Cre) at 2 × 10⁹ pfu and these were compared to Ad-5CMVeGFP (Ctrl). mTECs transfected with Cre show reduction in E-cadherin as assessed by **a** mRNA expression of *Cdh1* (encodes for E-cadherin), and **b** western blotting (representative image – left panel, and quantification – right panel). **c** Epithelial resistance was decreased, and **d** cellular velocity was increased in mTECs with E-cadherin knockdown. Data is expressed as median bars and generated of cells derived from 12 mice, 4 to 12 inserts. Normal human bronchial epithelial cells at ALI (normal controls) were transfected with Ad-GFP-U6-h-CFL1-shRNA (shCDH1) at 1.5 × 10⁹ pfu to knock down E-cadherin and these were compared to control adenovirus (Ad-GFP-U6-shRNA, GFP). Normal control cells transfected with shCDH1 show reduced **e** mRNA of CDH1 (encodes for E-cadherin) and **f** protein expression of E-cadherin (representative blot—left panel, and quantified blot—right panel). Assessment of the epithelial barrier function indicates that **g** monolayer resistance was decreased, **h** a trend towards increased permeability, and **i** cellular velocity was increased in control cells with knockdown of E-cadherin. Data is expressed as median bars and representative of 5 to 10 inserts per condition. Mann-Whitney test was performed and $P < 0.05$ was considered statistically significant.

**Overexpression of E-cadherin and Nrf2 activator restores epithelial function in COPD epithelia (COPD CELLS), and cigarette smoke (CS) injured epithelia.** We further evaluated if restoring E-cadherin in COPD CELLSs) can restore epithelial function. Increasing the E-cadherin expression in COPD CELLSs with adeno-CDH1 restored the barrier function and decreased cell velocity (Fig. 7a–c). However, not surprisingly, restoring E-cadherin did not have any effect on % moving and CBF (Fig. S8). To evaluate if overexpression of E-cadherin alters other cell adhesion proteins, we screened for the mRNA expression of claudins 1, 3, 7, and 10 (*CLDN1, CLDN3, CLDN7, CLDN8*, and *CLDN10*), occludin (*OCLN*), and tight junction proteins (*TJP1*, and *TJP2*) (Fig. 7d–k). COPD cells showed decreased mRNA expression of *CLDN1, CLDN8* and *TJP1*, and increased expression of *CLDN10* compared to healthy cells, but the differences were not statistically significant for *CLDN1*, and *CLDN8*. With overexpression of *CDH1* in COPD cells, the mRNA expression of *CLDN1, CLDN8*, and *TJP1* was increased, and *CLDN10* was decreased (Fig. 7d, g, and j).

We also observed that overexpression of E-cadherin in mTECs protects against CS-induced barrier disruption (Fig. 7l) and increased cell velocity (Fig. 7m). Moreover, E-cadherin overexpression maintained *Cdh1* mRNA expression despite CS exposure (Fig. 7n). Again, changes in E-cadherin expression did not alter % pixels moving and CBF (Fig. S9).

We have previously identified that Nrf2 pathway activators increase E-cadherin levels[26]. Therefore, as proof of principle, we treated the COPD cells and CS-exposed normal cells with 2-Cyano-3,12-dioxooleana-1,9(11)-dien-28-oic acid methyl ester (CDDO-Me), a potent Nrf2 activator to assess the effects of the pharmacological increase in E-cadherin. The CDDO-Me treated COPD cells significantly improved barrier function, decreases cell velocity, and increased CDH1 mRNA expression (Fig. 7o–q). Similarly, with CS injured epithelium, the CDDO-Me treatment significantly improved barrier function decreases cell movement

and protected against CDH1 mRNA downregulation induced by CS (Fig. 7r–t). However, CDDO-Me had no protection against loss of ciliated cells as quantified by % pixels moving and CBF in both COPD and CS-injured epithelia (Fig. S10).

**Overexpression of E-cadherin in AT2 cells decreases emphysema.** We further evaluated if overexpression of E-cadherin in AT2 cells protects against emphysema against elastase-induced injury. Elastase instillation caused emphysema in wild-type mice as measured by total lung capacity, residual volume, and compliance (Fig. 8a–c). However, *Cdh1* overexpression in AT2 cells protected the mice from elastase-induced damage (Fig. 8a–c). In addition, Cdh1 overexpressed in AT2 cells attenuated the elastase-induced airspace enlargement (Fig. 8d, e).

**Discussion**
In this study, we have uncovered the consequence of cell-specific inactivation of E-cadherin in lung epithelium in vivo. We found that loss of E-cadherin results in cell-specific tissue remodeling in the adult lung and that the loss of E-cadherin is sufficient to cause pathologic tissue remodeling. Using mouse and human in vivo, ex vivo, and in vitro models, we observed that loss of E-cadherin in ciliated cells induces barrier dysfunction in the monolayer (in vitro) and increases airway hyperresponsiveness (in vivo). As murine models do not undergo the airway remodeling observed in patients with COPD, further extrapolation is difficult. Still, airway hyperresponsiveness can be seen in some COPD patients with asthma-like features. Moreover, we identified that loss of E-cadherin reduces the proliferative capabilities of epithelial cells. As AT2 epithelial cell regeneration is critical in maintaining lung architecture[27,28], loss of E-cadherin in this model caused emphysema, which although not identical to the emphysema histologically present in patients with COPD, is indicative of local tissue destruction.

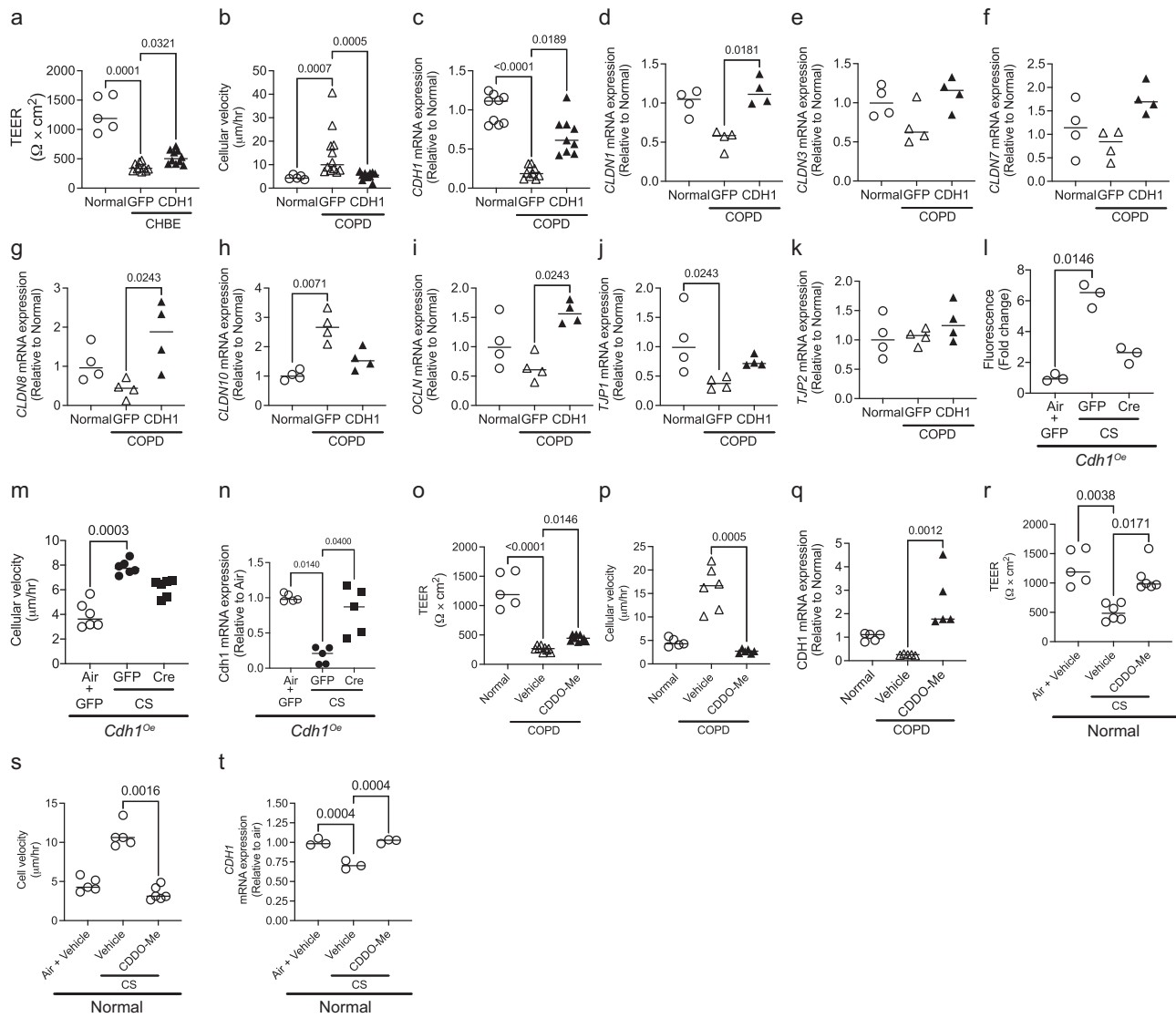

**Fig. 7 In vitro overexpression of E-cadherin or by activating the Nrf2 pathway protects E-cadherin expression and restores epithelial function in COPD and cigarette smoke (CS) injured epithelia.** COPD cells at 4 to 6 weeks ALI were transfected Ad-GFP-U6-h-CDH1 (CDH1) to overexpress E-cadherin or Ad-GFP (GFP) as control at 2 × 10⁹ pfu. COPD cells transfected with Ad-CDH1 result in **a** increased epithelial resistance, **b** reduced cellular velocity of COPD cells, and **c** increased mRNA expression of *CDH1*. Overexpression of E-cadherin in COPD cells **d**–**g** increased mRNA expression of claudins—*CLDN1*, *CLDN3*, *CLDN7*, and *CLDN8*, **h** decreased mRNA expression of CLDN10, **i** increased mRNA expression of occludin (OCLN), **j** increased mRNA expression of tight junction protein 1 (TJP1), and **k** TJP2 was not altered. Data is expressed as median bars and generated from 4 to 12 transwells per condition from two donors. To induce the over-expression of E-cadherin in Cdh1 knock-in mice, mice tracheal epithelial cells (mTECs) were transfected with adeno-Cre (Ad5CMVCre-eGFP) at 2 × 10⁹ pfu and were exposed to CS for 10 days. Overexpression of E-cadherin in mTECs protects against CS-induced epithelial functional phenotypes by **l** decreasing monolayer permeability, **m** decreasing the cellular velocity, and **n** protecting *Cdh1* mRNA downregulation due to CS exposure. Data for **l**–**m** involves three to six transwells per condition. COPD cells treated with CDDO-Me restore epithelial function by **o** improving the epithelial resistance, **p** decreasing the cellular velocity, and **q** increasing the *CDH1* mRNA expression of COPD cells as compared to age and gender-matched non-diseased epithelium (normal controls). Similarly, cigarette smoke (CS) exposed to healthy normal cells treated with CDDO-Me **r** restores epithelial resistance, **s** decreases cellular velocity, and **t** protects against *CDH1* mRNA downregulation due to CS exposure. (Data for **o**–**t** is generated from 3 to 6 transwells from 2 donors). For the panels the same data for normal are used for **a**–**k**, and **o**–**t** in the figure. Data are expressed as median bars. Kruskal-Wallis test, followed by Dunn's multiple comparison test was performed. *P* < 0.05 were considered statistically significant.

Our findings of E-cadherin loss of AT1 cells are particularly intriguing. As AT1 cells cover most of the alveolar surface[29]–[31], one could hypothesize that loss of E-cadherin would cause complete alveolar disruption with overwhelming injury. However, these mice were surprisingly minimally affected, raising the question of the relative contribution of E-cadherin to a mature alveolar monolayer. Furthermore, when AT1 epithelial cells are injured, AT2 epithelial cells undergo proliferation and differentiate into AT1 cells to regenerate the alveolar surface[32]. The data lead us to conclude that

E-cadherin plays a more prominent role in this regenerating epithelium and that temporary barrier disruption in the alveoli occurs with the loss of E-cadherin in AT1 cells and does not result in long-lasting histologic changes. This lack of alveolar destruction is likely critical as cell turnover is preserved with an intact AT2 epithelial cell. This is further emphasized by the fact that lung E-cadherin levels are restored by the sacrifice endpoint.

However, we do observe that knockdown of E-cadherin in AT1 epithelial cells increases mRNA expression of markers of

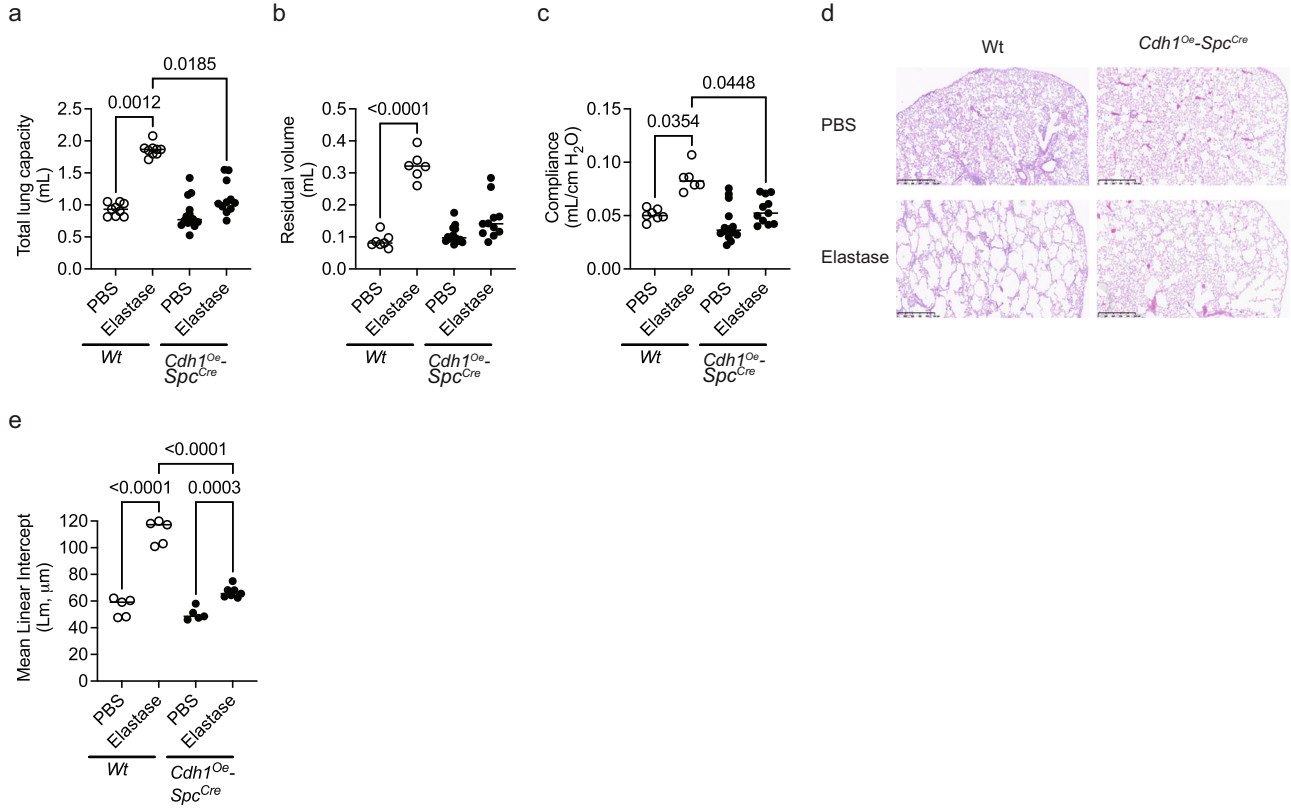

**Fig. 8 Overexpression of E-cadherin in AT2 protects against elastase-induced injury.** No difference in **a** total lung capacity, **b** residual volume, and **c** compliance among mice overexpressing E-cadherin in AT2 cells ($Cdh1^{Oe}$-$Spc^{Cre}$) instilled with elastase as compared to $Cdh1^{Oe}$-$Spc^{Cre}$ with PBS and wild-type (Wt) instilled with elastase. **d** H&E staining (representative image) and **e** quantitative MLI showed a reduction in airspace enlargement as compared to $Cdh1^{Oe}$-$Spc^{Cre}$ instilled with PBS and Wt instilled with elastase. Data are expressed as median bars and generated from 5 to 14 mice. Kruskal-Wallis test, followed by Dunn's multiple comparison test was performed. $P < 0.05$ were considered statistically significant.

epithelial to mesenchymal transition (EMT), a finding not seen with knockdown of E-cadherin in AT2 epithelial cells. There are several interesting points about the role of EMTs in COPD. First, EMT can result in fibrosis, and these markers, coupled with the increase in type III collagen and the restrictive lung defects identified in the mouse model, suggest fibrosis with a knockdown in AT1 cells. While these pathologic changes cannot be tied to a specific disease process, peribronchial fibrosis can be seen in COPD. Most studies indicated that loss of E-cadherin and concomitant upregulation of N-cadherin (neural cadherin) expression is the hallmark of EMT, though whether loss of E-cadherin is necessary and sufficient to induce EMT has been of debate[33,34]. While we[15] and other investigators[16,35–37] have identified markers of EMT in cigarette smoke-exposed and COPD tissues, these studies have been in the context of measuring these markers in mixed populations, where the cumulative effect of loss of E-cadherin in multiple epithelial cell types is measured.

Moreover, our lab has shown that tobacco smoke, even when inducing markers of EMT, can result in a cell behavior distinct from EMT and indicative of collective motion[15]. To our knowledge, we are the first to define a causal, cell-specific role for E-cadherin loss in chronic lung disease, and decreases in E-cadherin may not initiate EMT in every epithelial subtype. Future studies are needed to dissect the mechanisms mediating the subtype-specific cellular consequences of the downregulation of E-cadherin. Nonetheless, even in the absence of induction of EMT, E-cadherin loss in AT2 cells leads to tissue destruction.

We used genetic models of E-cadherin knockdown to study the consequences on tissue function and morphology. While critical in determining causality between loss of E-cadherin and tissue

remodeling, such a strategy cannot replicate the pathology that occurs with tobacco-associated COPD. Tobacco smoke is a complex mixture of several chemical compounds, including carbon monoxide, hydrogen cyanide, benzene, formaldehyde, nicotine, phenol, polycyclic aromatic hydrocarbons, tobacco-specific nitrosamines as well as particulates[38,39]. The regional distributions of these toxins and the chronicity of exposure cannot be replicated in a genetic model with a temporally limited, synchronized hit to specified cells. Furthermore, even with tobacco exposure, the mouse lungs do not result in the centrilobular emphysema seen in patients with COPD but exhibit patchy emphysema[40]. As such, it is impractical to expect that a genetic model of injury can result in the disease distribution seen in patients with COPD. However, we can extrapolate that whatever the cause, whether it be from regionalized toxicities from inhaled insults or more diffuse chronic damage, as seen in the aging lung, the regional decrease in E-cadherin in the epithelium can result in tissue remodeling that occurs in COPD.

Furthermore, strategies to upregulate E-cadherin improves cellular regeneration and barrier integrity and abrogate the development of emphysema. Current lung treatments focus on symptomatic improvement in patients with well-established lung disease. However, these data suggest a possible role for targeting E-cadherin in a vulnerable population as an early therapeutic strategy.

## Methods
**Animal and study design**. The study was approved by the Institutional Animal Care and Use Committee (IACUC) of the Johns Hopkins University Animal Use and Care Committee and complied with the Guidelines for Care and Use of

Laboratory Animals issued by the USA National Institute of Health. The study consisted of wildtype C57BL/6J (referred to as *Wt*), *Cdh1*$^{fl/fl}$, *Ager-CreER*$^{T2}$, *Sftpc-CreER*$^{T2}$, *Scbg1-CreER*$^{TM}$, and *Foxj1-CreER*$^{T2}$ mice. The C57BL/6J (Strain #000664), *Ager-CreER*$^{T2}$ (Strain #032771), *Scbg1-CreER*$^{TM}$ (Strain #016225), and *Foxj1-CreER*$^{T2}$ (Strain #027012) mice were purchased from the Jackson Laboratory (Bar Harbor, ME). In addition, we obtained the mating pair for *Cdh1*$^{fl/fl}$ mice from Dr. Andrew J. Ewald (Department of Cell Biology and Oncology, Center for Cell Dynamics, Sidney Kimmel Comprehensive Cancer Center, Johns Hopkins School of Medicine, Baltimore, USA)[41]. The mating pair for *Sftpc-CreER*$^{T2}$ mice (MGI ID: 5444645) was provided by Dr. Harold A Chapman (Lung Biology Center, UCSF, San Francisco)[42]. All mice were bred and maintained in a specific pathogen-free environment.

To generate E-cadherin-specific knockout in alveolar type I cells, *Cdh1*$^{fl/fl}$ mice were crossed with *Ager-CreER*$^{T2}$ mice to generate tamoxifen responsive *Cdh1*$^{fl/fl}$(*Ager-CreER*$^{T2}$) litters (referred to as *Cdh1*$^{fl/fl}$-*AgerCre*). To generate E-cadherin-specific knockout in alveolar type II cells, *Cdh1*$^{fl/fl}$ were crossed with *Sftpc-CreER*$^{T2}$ mice to generate tamoxifen responsive *Cdh1*$^{fl/fl}$ (*Sftpc-CreER*$^{T2}$) litters (referred to as *Cdh1*$^{fl/fl}$-*SpcCre*). To generate E-cadherin-specific knockout in ciliated cells, *Cdh1*$^{fl/fl}$ were crossed with *Foxj1-CreER*$^{T2}$ mice to generate tamoxifen-responsive *Cdh1*$^{fl/fl}$(*Foxj1-CreER*$^{T2}$) litters (referred to as *Cdh1*$^{fl/fl}$-*Foxj1Cre*). To generate E-cadherin-specific knockout in club cells, *Cdh1*$^{fl/fl}$ were crossed with *Scbg1-CreER*$^{TM}$ mice, to generate tamoxifen responsive *Cdh1*$^{fl/fl}$ (*Scbg1-CreER*$^{TM}$) litters (referred as *Cdh1*$^{fl/fl}$-*Scbg1a1Cre*).

We have created a mouse *Cdh1* conditional knock-in (gene for E-cadherin, which will be referred to as *Cdh1*$^{Oe}$) at the locus of ROSA26 in C57BL/6 mice by CRISPR/Cas-mediated genome engineering. The CAG-loxP-Stop-loxP-Kozak-mouse Cdh1 CDS-polyA cassette was cloned into intron 1 of ROSA26 in reverse orientation. The expression of mouse *Cdh1* is dependent on Cre recombinase.

To generate E-cadherin-specific knock-in in alveolar type II cells, *Cdh1*$^{Oe}$ were crossed with *Sftpc-CreER*$^{T2}$ mice to generate tamoxifen responsive *Cdh1*$^{Oe}$ (*Sftpc-CreER*$^{T2}$) litters (referred to as *Cdh1*$^{Oe}$-*SpcCre*). The mice were genotyped according to the optimized PCR primers described in Table S1.

**Intratracheal instillation to knock down E-cadherin in mice lung**. *Cdh1*$^{fl/fl}$ mice were anesthetized with ketamine (100 mg/kg) - xylazine (15 mg/kg) solution via intraperitoneal injection. To knock out E-cadherin from *Cdh1*$^{fl/fl}$ mice, lung instillation of either Adenovirus Ad5CMVeGFP (Control, Ctrl) and Ad5CMVCre-eGFP (Cre) at a titer of $2 \times 10^9$ pfu in 60 μL was performed as described previously on 10th day for one month or 2 months or 3 months[43,44].

**Intratracheal elastase administration**. *Cdh1*$^{Oe}$-*SpcCre* and *Wt* mice were anesthetized with ketamine (100 mg/kg) - xylazine (15 mg/kg) solution via intraperitoneal injection. A single intra-tracheal dose of six enzymatic activity units (U) of porcine pancreatic elastase (EC-134, Elastin Products, MO, USA) dissolved in 50 μL of 1X PBS was delivered as described previously[44]. The control group received 50 μL of 1X PBS only.

**Tamoxifen administration**. At 5 weeks of age, mice were switched to either a tamoxifen chow diet (TD.130856, Envigo, IN, USA) or a normal chow diet (Envigo, IN, USA) for 31 days.

**Physiological measurements**. Mice were anesthetized with ketamine (100 mg/kg) - xylazine (15 mg/kg) solution. Once sedated, the trachea was cannulated with an 18-gauge needle. Electronic controls on the ventilator are set to periodically lengthen the inspiratory time to measure total lung capacity, residual volume, and lung compliance[44]. The *Cdh1*$^{fl/fl}$-*Foxj1*$_{Het/WT}$$^{Cre}$ and *Cdh1*$^{fl/fl}$-*Scbg1a1*$^{Cre}$ mice were further given stepwise increases (0, 3, and 30 mg/mL) of Acetyl- β-methylcholine chloride (Sigma Aldrich, MO, USA), to assess airway reactivity[45].

**Lung fixation and histopathology**. After performing physiological measurements, the lungs were fixed by inflation with 25cmH$_2$O with instilled 10% buffered formalin (Avantik, NJ, USA), and fixed lung volume was assessed as described previously by us[43]. In addition, the fixed lungs were embedded in paraffin, and sagittal sections were stained with hematoxylin and eosin (H&E) or Masson's Trichrome at Reference Histology Laboratory, Johns Hopkins Medical Institutions—Pathology (Baltimore, USA).

**Human airway epithelial cell culture**. Primary non-diseased human bronchial epithelial (normal) and COPD human bronchial epithelial (COPD) cells were purchased from MatTek Life Sciences (Ashland, MA, USA) or Lonza (Basel, Switzerland) were expanded on collagen-coated T$_{75}$ flask and differentiated into pseudostratified epithelium at the air-liquid interface (ALI) on Transwells with spolyester membranes with 0.4 μm pores as described by us before[4,14,15]. Normal and COPD cells were also grown in four Chamber Cell Culture Slides, Sterile (CELLTREAT Scientific Products, MA, USA) for 24 h and were not differentiated at ALI.

**E-cadherin knockdown in healthy controls and overexpression in COPD cells**. Adenovirus to knockout E-cadherin (Ad-GFP-U6-h-CDH1-shRNA, shCDH1), Adenovirus to overexpress E-cadherin (Ad-GFP-U6-h-CDH1, CDH1), and Control Adenovirus with GFP (Ad-GFP, GFP), were purchased from Vector Biolabs (Malvern, PA, USA) to transduce cells. At chamber slides, normal control cells were transduced with shCDH1 to knock down E-cadherin, and COPD cells were transduced with CDH1 to overexpress E-cadherin at $0.5 \times 10^9$ pfu in 500 μL. These were compared to respective control cells or COPD cells transfected with GFP.

Well-differentiated patient-derived normal cells at 4 to 6 weeks ALI were treated apically and basolateral with transfection media (containing PneumaCult$^{TM}$-ALI medium and 5 μg mL$^{-1}$ polybrene with Ad-GFP-U6-h-CDH1-shRNA or Ad-GFP at $1.5 \times 10^9$ pfu in 1300 μL) for 4 h. After 4 h, the transfection media was transferred from the apical surface to the basolateral. After 20 h, the transfecting media was replaced with PneumaCult$^{TM}$-ALI. The culture was maintained for additional 48 h. Post 48 h, the epithelium at ALI was visualized for Green Fluorescent Protein (GFP) at 20× by fluorescent microscopy using 3i Marianas/Yokogawa Spinning Disk Confocal Microscope (Leica Microsystems, TX, USA), assessed for functional phenotypes (as described below), and samples were collected for western blot assay.

**E-cadherin overexpression in COPD CELLSs**. Adenovirus to overexpress E-cadherin (Ad-GFP-U6-h-CDH1) and control adenoviruses (Ad-GFP), was purchased from Vector Biolabs (Malvern, PA, USA) to transduce COPD cells at 4–6 weeks ALI at $2 \times 10^9$ pfu in 1300 μL as described in the earlier section (*E-cadherin knockdown in normal cells and overexpression in COPD cells*).

**Isolation of mice tracheal epithelial cells (mTECs)**. The *Cdh1*$^{fl/fl}$ mice were euthanized following Carbon dioxide narcosis followed by cervical dislocation IACUC guidelines. The tissue-free lumen exposed trachea dissected from the 10 *Cdh1*$^{fl/fl}$ mice were added to 10 mL of 1X-Phosphate Buffer Saline (1X-PBS, ThermoFisher Scientific, New York, USA) supplemented with Penicillin–Streptomycin (ThermoFisher Scientific, NY, USA). The trachea was transferred to 15 mL 0.15% Pronase solution and incubated overnight at 4 °C. The tube was rocked 40–50 times and was passed through a 70 μm cell strainer (Corning Life Sciences, MA, USA). The cell strainer consists of digested trachea and the solution consists of cells. The cell solution was topped up to a volume of 50 mL with1X-PBS and centrifuged at 300 g with the brake on. The resulting pellet was suspended with 15 mL of DMEM supplemented with 10% Fetal Bovine Serum (FBS) and Penicillin-Streptomycin and incubated at 37 °C and 5% CO$_2$ for 4 h in a T$_{75}$ flask (Corning Life Sciences, MA, USA) to eliminate fibroblasts and mononuclear cells (as fibroblasts and mononuclear cells get to adhere to flask). After 4 h, the cells were suspended in epithelial expansion media (Prepared by mixing 490 mL PneumaCult™-Ex Plus Basal Medium Supplemented with 10 mL PneumaCult™-Ex Plus 50X Supplement, 0.5 mL Hydrocortisone stock solution, and 1% Penicillin–Streptomycin: StemCell Technologies Inc., Vancouver, Canada). The suspended cells are expanded in a T$_{75}$ collagen-coated flask at 80–90% confluency. The cells were plated onto Rat tail collagen I coated 0.4 μm pore polyester membrane 12 mm Transwell® at a seeding density of 300,000 cells/well with epithelial expansion media on apical and basolateral surfaces. At 100% confluency, the transwells were put at air–liquid interface (ALI) media (Prepared by 450 mL PneumaCult™-ALI medium supplemented with 50 mL of PneumaCult™-ALI 10X Supplement, 5 ×1 mL vial of 100× PneumaCult™-ALI Maintenance Supplement, 1 mL Hydrocortisone Solution, 1 mL Heparin Solution and 1% Penicillin-Streptomycin; StemCell Technologies Inc., Vancouver, Canada) on the basolateral surface for 2 weeks to obtain a fully differentiated epithelium.

**In vitro knockdown of E-cadherin in mTECs**. The mTECs of *Cdh1*$^{fl/fl}$ mice at two weeks ALI were transfected with either Adenovirus Ad-5CMVeGFP (Control) or Ad5CMVCre-eGFP (Cre) at a titer of $2 \times 10^9$ pfu in 1300 μL with transfection protocol as described in an earlier section (*E-cadherin knockdown in normal control cells and overexpression in COPD cells*).

**Over-expression of E-cadherin in mTECs**. The mTECs of *Cdh1* conditional knock-in mice at two weeks ALI were transfected with either Adenovirus Ad-5CMVeGFP (Control) or Ad5CMVCre-eGFP (Cre) at a titer of $2 \times 10^9$ pfu in 1300 μL with transfection protocol as described in an earlier section (*E-cadherin knockdown in NORMALs and overexpression in COPD cells*).

**Cigarette smoke (CS) exposure to normal controls cells**. The control cells at 4–6 weeks ALI were exposed to either exposed CS smoke or humidified air for 10 days as mentioned previously by us[4,14,15]. One CS exposure consisted of two cigarettes that burned for ~8 min using the ISO puff regimen.

**Treatment of Nrf2-pathway activator**. In this study CDDO-Me (Toronto Research Chemicals) a potent Nrf2-pathway activator at a dose of 100 nM was utilized. The basolateral treatment with CDDO-Me was 5 days for COPD cells and 10 days for the normal control cells at ALI exposed to CS.

**Quantification of epithelial functional phenotypes**. The epithelial functional phenotypes were assessed by quantifying the barrier function, paracellular permeability, ciliary beat frequency (CBF), % moving cilia, and cellular velocity as described below.

**In vitro barrier function assessment**. To determine the epithelial integrity of the human and mice pseudostratified monolayer epithelium at ALI, the trans-epithelial electrical resistance (TEER) was measured using an epithelial voltohmeter (EVOM, World Precision Instruments Inc, FL, USA) with the STX2 electrodes as previously described[4,14,15]. Values were corrected for fluid resistance (insert with no cells) and surface area.

**Paracellular permeability**. The paracellular permeability of the epithelium at ALI was determined using fluorescein isothiocyanate-dextran (FITC-Dextran) flux assay as described previously[4,15].

**CBF**. The epithelium at ALI was incubated at 37 °C and 5% $CO_2$ in the 3i Marianis/Yokogawa Spinning Disk Confocal Microscope (Leica Microsystems, TX, USA), and high-speed time-lapse videos were taken at 32× air at 100 Hz with a total of 250 frames using a scientific Hamamatsu C1140-42U30 CMOS camera (Hamamatsu Photonics, NJ, USA) as reported in our previous publications[14,15]. Five areas were imaged per insert. A Matlab (R2020a) script (validated against SAVA[46]) was used to determine average CBF and % moving cilia per video to generate a heatmap[47].

**Cellular velocity**. Cellular velocity was quantified by performing Particle Image Velocimetry (PIVlab) on Matlab (R2020a), using multi-pass cross-correlation analysis with decreasing interrogation window size on image pairs to obtain the spatial velocity as described previously by us[48]. Using phase-contrast microscopy at 3i Marianis/Yokogawa Spinning Disk Confocal Microscope at 32×, and 37 °C and 5% $CO_2$ incubation, we captured time-lapse videos of the pseudostratified monolayer epithelium every 5 min for 2 h, and average velocity was computed for the area. Five areas were imaged per insert.

**Western blot assay**. Equal quantities of protein were separated on NuPAGE™ 4–12%, Bis-Tris gradient gel (ThermoFisher Scientific, NY, USA) using a Mini Gel Tank (ThermoFisher Scientific, New York, USA) at 200 V for 30 min. Proteins were transferred to an Immobilon-P PVDF membrane (Millipore Sigma, MA, USA) at 47 V overnight at 4 °C using Mini Trans-Blot® Cell (Bio-Rad Laboratories, CA, USA). Following the transfer, the blots were blocked with 5% w/v of Non-Fat Powdered Milk (Boston BioProducts, MA, USA) in 1X PBS with 0.1% Tween® 20 Detergent, (1X PBST, Millipore Sigma, MA, USA) for 2 h at room temperature on a shaker. The blocked membrane was washed with 1X PBST and then incubated in primary antibody (E-cadherin (24E10) Rabbit mAb, 135 kDa and GAPDH (14C10) Rabbit mAb, 37 kDa antibodies from Cell Signaling Technology, MA, USA) at a concentration of 1:1000 overnight at 4 °C. The blot was further washed with 1X PBST and then incubated in secondary antibody IRDye® 800CW Goat anti-Rabbit (LI-COR, NE, USA) at a dilution of 1:10,000 for 1 h at room temperature. Blots were washed three times with 1× PBST, then visualized on a Li-Cor imager. Images were analyzed using Fiji[49].

**Preparation of samples for immunofluorescence**. The lung tissue sections were prepared for immunofluorescent staining by performing deparaffinization, rehydration, and antigen unmasking as described by Crosby et al.[50].

Normal control and COPD cells at weeks 1 to 3 ALI were fixed using 4% Paraformaldehyde solution in PBS (ThermoFisher Scientific, MA, USA). Sucrose infiltration steps were performed on normal control and COPD cells[15], followed by embedding in a biopsy-size cryomold using Optimal Cutting Temperature (O.C.T) compound (Tissue-Tek, CA, USA) and cutting them to 10 µm sections using Leica Cryostat CM3050 S (Leica Biosystems Inc., IL, USA). The sections were attached to Superfrost Plus microscopic slides (Fisher Scientific, PA, USA) and dried at room temperature overnight.

**Immunofluorescence assay**. The sections were permeabilized with 0.1% Triton™ X-100 (Millipore Sigma, MO, USA) in 1X PBS and blocked for 2 h in 1X PBS with goat serum (10%) (ThermoFisher Scientific, MA, USA). The primary antibody used in this study were Recombinant Anti-Prosurfactant Protein C antibody [EPR19839] (dilution of 1:100, ab211326, Abcam, MA, USA), Anti-BrdU antibody [IIB5] (5-Bromo-2'-deoxyuridine, Thymidine analog, dilution of 1:100, ab8152, Abcam, MA, USA), β-tubulin (D2N5G) Rabbit mAb #15115 (dilution of 1:200, Cell Signaling Technology, MA, USA), Cytokeratin 14 (LL001) (dilution of 1:100, sc-53253, Santa Cruz Biotechnology, Inc., TX, USA), E-cadherin (24E10) Rabbit mAb #3195 (dilution of 1:200, Cell Signaling Technology, MA, USA), E-cadherin (4A2) Mouse mAb #14472 (dilution of 1:100, Cell Signaling Technology, MA, USA), and Recombinant Anti-Mucin 5AC antibody [45MI] (dilution of 1:50, ab3649, Abcam, MA, USA), diluted in blocking solution, was incubated on the tissue slides overnight at 4 °C. The tissue sections were further washed with 1X PBS and incubated with Goat anti-Rabbit IgG (H + L) or Goat anti-Mouse (H + L) Cross-Adsorbed Secondary Antibody - Alexa Fluor 488 (Rabbit - A11034, Mouse - A11001) / Alexa

Fluor 555 (Rabbit - A21428)/Alexa Fluor 647 (Rabbit - A21244, Mouse -A21236) (ThermoFisher Scientific, MA, USA) diluted at 1:500 for 1 h at room temperature. The tissue was then stained with 2 µg mL$^{-1}$ of Hoechst 33342 (ThermoFisher Scientific, MA, USA) for 15 min at room temperature, washed, and mounted with Prolong™ Gold Antifade Mountant (ThermoFisher Scientific, MA, USA). The images were then captured in a Zeiss LSM700 Confocal microscope.

**Quantitative polymerase chain reaction (qPCR) assay**. Total RNA was isolated from cultured human bronchial epithelial cells/mice right lung tissues and purified using the Qiagen AllPrep DNA/RNA Mini Kit (Qiagen, Hilden, Germany), supplemented with the Proteinase K (Qiagen, Hilden, Germany). cDNA of 1000 ng µL$^{-1}$ was obtained using the High-Capacity cDNA Reverse Transcription Kit (ThermoFisher Scientific, MA, USA), and the absence of DNA contamination was verified by excluding the reverse transcriptase from subsequent PCR reaction. cDNA was subjected to PCR using the SYBR™ Green PCR Master Mix (ThermoFisher Scientific, MA, USA) to amplify the epithelial, mesenchymal, or fibrosis markers listed in Supplementary Table S2.

Each PCR reaction was carried out as follows: initial denaturation of 94 °C for 15 min, 45 cycles of 94 °C for 35 s, 60 °C for 1 min, and 72 °C for 1 min 15 s, followed by a final extension at 72 °C for 2 min. Each cycle was repeated 45 times. Based on the comparative Ct method, gene expression levels were calculated, and GAPDH was used as the housekeeping gene.

**Quantification of mean linear intercept**. To quantify the changes in lung structure, we calculated the mean linear intercept as described previously using Stepanizer[44,51].

**Quantification of intensity**. The intensity of immunofluorescence images was analyzed using Fiji[49]. Images were converted to greyscale and then threshold to create a mask defining the region of interest. The intensity of the areas indicated by the mask was then measured.

**Statistics and reproducibility**. Prism version 9.0 (GraphPad, CA, USA) and Matlab (R2020a) were used to analyze the data. The results are expressed as a median. To compare the data sets involving two groups, the Mann-Whitney test was performed. To compare more than two groups, the Kruskal-Wallis test, followed by Dunn's multiple comparison test was performed.

**Reporting summary**. Further information on research design is available in the Nature Research Reporting Summary linked to this article.

## Data availability
All data supporting this study are available within the article and its Supplementary Information. The uncropped and unedited blots are presented in Fig. S11. Source data are available within the Supplementary Data 1 and Supplementary Data 2 file. All other data are available upon submission of request to the corresponding author.

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

## Acknowledgements

We thank Johns Hopkins University (JHU)—School of Medicine (SoM) Microscope Facility for providing access and training in the 3i Marianas Spinning Disk Confocal and Zeiss LSM700 single-point laser scanning confocal microscope. We would also like to thank JHU-SoM Reference Histology Laboratory for assisting in mice lung tissue's H&E and Masson trichrome staining. We would also like to thank JHU-SoM Tumor Micro-environment Laboratory for providing access and training in the Hamamatsu Nano-Zoomer brightfield slide scanner. The Research reported in this publication was supported by the National Heart, Lung, and Blood Institute (NHLBI R01-HL124099 and HLR01-HL151107) and the Office of the Director of the National Institutes of Health under award number S10 OD016374 (to Scot Kuo of the JHU Microscope Facility). V.K.S. is a member of the Johns Hopkins Center for Cell Dynamics.

## Author contributions

A.W., B.G., B.Y.L., C.S., D.B., D.C., E.C., E.T., K.N., L.Y., M.B., M.G., M.Z., P.P.C., S.C., S.M., and S.T. executed all experiments. BG contributed to the experimental design, implementation, prepared figures, data analysis, interpretation, and manuscript drafting. J.L. and H.R. performed the lung function test on mice under the supervision of W.M. S.B. contributed to editing the manuscript. V.K.S. is the principal investigator, who conceived the idea, led the study design and data interpretation, and edited the manuscript. All authors have read and approved the manuscript.

## Competing interests

The authors declare no competing interests.
