## [Peer Review File · Communications Biology]

Reviewers' comments:

Reviewer #1 (Remarks to the Author):

1. The authors have undertaken an extensive study on in-vitro cultured lung epithelial cells and a mouse model. They have been able to modulate the expression on the inter-cell adherens protein E-cadherin, and have used this to investigate the physiological and potential pathological effects of E-cad down regulation. The authors have done a great deal of what looks like careful and skilled laboratory work and report a prodigious and indeed rather overwhelming amount of data. I would like to see a reordering of the data to better bring out what I think are the messages and also to tidy up some of the pathogenic loose-ends. I will try and develop what I mean by this to help this process along.

2. Firstly, the authors need to understand the firm messages coming out of research in COPD over the last 10 years (building on work going back to the 1970s), although I have to accept that these are slow to be widely transmitted (the emphasis is still on luminal events of infections and innate immune activation). Smoking-related COPD is essentially excessive and progressive airflow obstruction due to fibrosis, narrowing and destruction of small airways associated with EMT activity, with related air-trapping. Gradually emphysema develops especially in these areas of air trapping; this is the tell-tale centri-lobular emphysema of most smokers, and quite different to the pan-acinar emphysema due to genetic proteinase deficiencies.

3. Now what I think the authors are saying is that saying is that down-regulation of E-cad increases epithelial monolayer leakiness and cell proliferation, leading to dilatation of small airways and pan-acinar emphysema, but not EMT activation or classic changes of smoking-related COPD at all. Indeed, these changes interestingly are much more like what is described in a pure ageing effect (and it might be interesting to know what is happening coincidentally to telomere length?). Paradoxically, the only evidence for EMT activity with lung fibrosis is when Type 1 Alveolar epithelial cells have their E-cad down-regulated; this is reminiscent of IPF but in this disease the proliferative and EMT activity seems in humans to be related especially to Type 2 cells.

4. So, there are some pretty interesting findings here, but in relation to human disease it is mostly rather aberrant; but that is very interesting in its own right, as is the separation of E-cad down-regulation from EMT activity for the most part. Thus, the down-regulation of E-cad with cigarette smoke (CS) exposure (accompanied by up-regulation of N-caderin etc) is quite different in outcomes to singular E-cad down-regulation as here. It seems to me that this is important. In general, a detailed comparison of the differential effects of CS versus manipulated E-cad down-regulation would be vital.

5. The Discussion in particular needs to be substantially strengthened to bring this narrative together in a way meaningful for their main purpose of understanding where E-cad stands in relation to human disease/ageing. I think the data they have can be interpreted in a much more robust, less speculative and sophisticated way. As I have said, amplifying this with current or previous comparative data on CS effects would strengthen it appreciably.

6. In general the document is quite readable, but these are places where the word choice is sub-optimal, e.g. line 57, "stimulated" would be better than "spurring"; line 58 "exposure of ...cells ...to..."; line 78-79 is quite garbled; in line 88 "in" for "on". As is frequently the case for non-English-speaking authors, the choice of prepositions, a/the, and singular/plural nouns is a bit problematic but a relative minor issue.

Reviewer #2 (Remarks to the Author):

Authors found the novel aspects of E-cad in lung epithelial cells and the defect of E-cad in various kind of cells resulted in emphysematous mimicking COPD phenotype. The most of the results were represented by morphometry, lung functions and the expression of E-cad. It seemed that E-cad can explain any kinds of pathogenesis in COPD, however, the decrease of E-cad expression in COPD lungs has been shown by various studies including ERGR activity, protease activity, EMT and inflammation. If we should be able to increase the expression of E-cad in lung as a novel treatment, it would be cleaved by disease associated factors in microenvironment in the long run. The importance of E-cad in COPD pathogenesis should be shown by systematic science, not by broad and superficial phenotypes.

There are the things should be concerned in the manuscript,

1. The mechanisms to induce airspace enlargement in every gene modified mice. Especially need to argue about the similarity to COPD pathogenesis.
2. What were the missing signal pathways when each specific cells lost E-cad signaling.
3. Authors should show the function of E-cad with novel findings.
4. When the AT1 was lost and airspace enlargement occurred, why AT2 cells could not compensate the damage. Please show the mechanism.
5. What were the lost homeostatic functions of AT2 cells, when E-cad was deleted. It looks that AT2 cell could not maintain itself.
6. Only club cell was not studies in this manuscript. Club cell in one of the most important cells in COPD pathogenesis. Its E-cad function is the key for the defense against external stimuli. Scgb1a1-CreERT2 promoter mouse is available.

E-cad deletion in different cell types resulted in different dysfunction in vivo. This is still the entrance of the study themes and can be enriched with more effort. I hope authors accumulate the more datasets and hope resubmit it to the journal.

Reviewer #3 (Remarks to the Author):

Previous studies from authors' and other laboratories showed that chronic obstructive pulmonary disease (COPD) involves down-regulation of E-cadherin, an adhesion molecule that is important for maintaining the epithelial architecture, suggesting that E-cadherin loss could be a primary cause of alveolar epithelium disruption occurring in this disease. The present study examined this possibility by conducting various experiments. Authors confirmed that genetic removal of E-cadherin in adult mice induces emphysema-like deformation of pulmonary alveoli, and additionally revealed that E-cadherin loss in Type-II alveolar cells, but not Type-I cells, is important for eliciting this abnormality. Furthermore, E-cadherin removal reduced the proliferation capacity of lung epithelial cells. The authors also examined the effect of E-cadherin overexpression in human bronchial epithelial cells derived from COPD patients, finding that this treatment could restore normal epithelial functions in these cells. Together with other experiments, the authors conclude that E-cadherin in Type-II cells plays a crucial role in maintaining epithelial architecture as well as its regenerative capacity in the lung.

These findings are not sort of surprising, because our knowledge about the role of E-cadherin in maintenance of epithelial integrity is well established. Nevertheless, to examine whether E-cadherin reduction in alveolar cells is causal to the emphysematous injuries is of significance, as other mechanisms might have primarily contributed to this epithelial defect in COPD. From this point of view, the authors have been successful at least in part, as they were able to confirm that E-cadherin plays a crucial role in epithelial maintenance in the adult lung. The finding that E-cadherin in Type-II alveolar cells solely serves for protecting the lung from emphysema genesis is novel and intriguing, although why its loss in Type-I cells shows only minor abnormalities remains unresolved.

I suggest a few minor points which should be improved before publication of this manuscript.

1. In testing the idea that E-cadherin reduction is causal to the emphysematous injuries, one of important experiments is to study whether overexpression of E-cadherin in COPD cells is sufficient to restore the normal epithelial structure and functions in these cells. To do this, authors used CS-exposed mTEC and CHEBEs, demonstrating significant recovery of their barrier function due to E-cadherin overexpression. However, they have never performed histological or cytological observations to look at what kinds of structural changes were induced in these cells. I suggest co-immunostaining of the cells for E-cadherin and a tight junction protein such as ZO-1, to confirm whether normal cell junctions, which had been lost in COPD cells, were morphologically restored by E-cadherin overexpression.

2. In some immunofluorescence images (e.g., Fig. 1 (1 month) and Fig. 3), it is very difficult to see whether E-cadherin level really differs between the control and experimental conditions. This problem should be addressed by providing better images or by adding a larger magnification to each of the image panels. To the Fig. 3B panels, the label 'BrdU' should be added.

3. Since immunofluorescence data are not so persuasive in general in this manuscript, quantification of fluorescence signals is necessary for each marker to convince readers of the authors' conclusions.

Reviewer 1 Comments for the Author:

The authors have undertaken an extensive study on in-vitro cultured lung epithelial cells and a mouse model. They have been able to modulate the expression on the inter-cell adherens protein E-cadherin, and have used this to investigate the physiological and potential pathological effects of E-cad down regulation. The authors have done a great deal of what looks like careful and skilled laboratory work and report a prodigious and indeed rather overwhelming amount of data. I would like to see a reordering of the data to better bring out what I think are the messages and also to tidy up some of the pathogenic loose ends. I will try and develop what I mean by this to help this process along.

We thank the reviewer for the thoughtful reading of the manuscript and appreciate that the reviewer recognized the overwhelming amount of data already present in the manuscript. We have significantly reorganized to address the concerns raised and appreciate the reviewer's insights.

Firstly, the authors need to understand the firm messages coming out of research in COPD over the last 10 years (building on work going back to the 1970s), although I have to accept that these are slow to be widely transmitted (the emphasis is still on luminal events of infections and innate immune activation). Smoking-related COPD is essentially excessive and progressive airflow obstruction due to fibrosis, narrowing and destruction of small airways associated with EMT activity, with related air-trapping. Gradually emphysema develops especially in these areas of air

trapping; this is the tell-tale centri-lobular emphysema of most smokers, and quite different to the pan-acinar emphysema due to genetic proteinase deficiencies.

Now what I think the authors are saying is that down-regulation of E-cad increases epithelial monolayer leakiness and cell proliferation, leading to dilatation of small airways and pan-acinar emphysema, but not EMT activation or classic changes of smoking-related COPD at all. Indeed, these changes interestingly are much more like what is described in a pure ageing effect (and it might be interesting to know what is happening coincidentally to telomere length?). Paradoxically, the only evidence for EMT activity with lung fibrosis is when Type 1 Alveolar epithelial cells have their E-cad down-regulated; this is reminiscent of IPF but in this disease, the proliferative and EMT activity seems in humans to be related especially to Type 2 cells.

We appreciate the reviewer for pointing out the subtlety of the findings. We are indeed saying that the downregulation of E-cadherin increases monolayer leakiness and decreases cell proliferation. And we find that this occurs with cell specificity, as do the findings of classic EMT activation. However, we respectfully disagree with the statement that changes are distinct from the classic changes of smoking-related COPD. We and others have repeatedly shown that cigarette smoke exposure decreases E-cadherin (Fig. S4, and PMID: 28642260 & 35118497). This manuscript aims to demonstrate the causal relationship between E-cadherin loss and tissue remodeling in COPD. Of course, the models used in this manuscript with uniform, cell-specific deletion in E-cadherin do not occur in tobacco-associated lung disease, where the regional flow dynamics dictate the degree and duration of cellular exposure to tobacco and therefore result in non-uniform changes in E-cadherin. But our study shows that a reduction in E-cadherin is sufficient to lead to lung pathology in COPD in that area. Moreover, while EMT has been demonstrated in COPD tissues, it cannot be extrapolated that every COPD epithelial cell that loses E-cadherin triggers this EMT process, only that the cumulative loss results in some evidence of EMT. Our studies indicate that this is a cell-specific process.

The Discussion in particular needs to be substantially strengthened to bring this narrative together in a meaningful for their main purpose of understanding where E-cad stands in relation to human disease/ageing. I think the data they have can be interpreted in a much more robust, less speculative and sophisticated way. As I have said, amplifying this with current or previous comparative data on CS effects would strengthen it appreciably.

We appreciate the reviewer's comments and have edited the discussion to better discuss these points.

In general, the document is quite readable, but these are places where the word choice is sub-optimal, e.g. line 57, "stimulated" would be better than "spurring"; line 58 "exposure of ...cells ...to..."; line 78-79 is quite garbled; in line 88 "in" fo "on". As is frequently the case for non-English-speaking authors, the choice of prepositions, a/the, and singular/plural nouns is a bit problematic but a relative minor issue.

We appreciate the reviewer's comments, and many of the errors that occurred were due to moving the text while editing the manuscript and have been corrected.

Reviewer 2 Advance Summary and Potential Significance to Field:

Reviewer 2 Comments for the Author:

Authors found the novel aspects of E-cad in lung epithelial cells and the defect of E-cad in various kind of cells resulted in emphysematous mimicking COPD phenotype. The most of the results were represented by morphometry, lung functions and the expression of E-cad. It seemed that E-cad can explain any kinds of pathogenesis in COPD, however, the decrease of E-cad expression in COPD lungs has been shown by various studies including ERGR activity, protease activity, EMT and inflammation. If we should be able to increase the expression of E-cad in lung as a novel treatment, it would be cleaved by disease associated factors in microenvironment in the long run. The importance of E-cad in COPD pathogenesis should be shown by systematic science, not by broad and superficial phenotypes.

We appreciate that the focus of the manuscript has been more on the phenotypic changes than the molecular mechanisms mediating changes. We agree that downstream mechanisms by which E-cadherin can alter cellular function have been demonstrated by both our group as well as others (PMID: 28642260, 35118497, 26930653, 10439038, 35531891, 25079037, 35531891). In addition, we have added to those findings, by identifying cell-specific mechanisms that result in disease, with the decreased barrier contributing to airway dysfunction, and reduced proliferation leading to emphysema. However, the studies in the literature have left us with the fundamental question of causality which no other group has not demonstrated to date, and in fact cannot be addressed without an extensive phenotypic assessment. Answering this question is of critical importance since it justifies trying to target this protein for therapeutic purposes. However, the reviewer does raise an important point of whether increasing E-cadherin in the lung can serve this therapeutic purpose. To address this question, we have now included data on the protection provided by E-cadherin overexpression in AT2 cells of mice lungs (Figure 8).

The mechanisms to induce airspace enlargement in every gene-modified mice. Especially need to argue about the similarity to COPD pathogenesis. What were the missing signal pathways when each specific cells lost E-cad signaling.

We appreciate the reviewer's comments and have highlighted the importance of AT2 proliferation in airspace enlargement and the protection that occurs with E-cadherin overexpression (Figure 8).

When the AT1 was lost and airspace enlargement occurred, why AT2 cells could not compensate the damage. Please show the mechanism.

We apologize for any confusion there was in the manuscript. The reviewer is correct that when E-cadherin in AT1 cells is lost, AT2 cells should compensate for the damage. This is what we have demonstrated, with E-cadherin knockdown in AT1 cells, AT2 cells compensate with no evidence of airspace enlargement, and only some fibrosis (Fig. 2D & E, S1 & S2). And with loss of E-cadherin in AT2 cells, we see airspace enlargement (Fig. 2I & J).

What were the lost homeostatic functions of AT2 cells, when E-cad was deleted. It looks that AT2 cell could not maintain itself.

This is correct, and we demonstrate that this occurs due to a defect in the proliferation of the cell, which is restored with overexpression of E-cadherin (Fig. 8). While further studies are needed to better delineate the contribution of E-cadherin to proliferation and regeneration, studies are ongoing and will require extensive molecular dissection, and therefore are out of the scope of this manuscript, which as Review 1 pointed out is already quite extensive.

Only club cell was not studies in this manuscript. Club cell in one of the most important cells in COPD pathogenesis. Its E-cad function is the key for the defense against external stimuli. Scgb1a1-CreERT2 promoter mouse is available.

We appreciate the reviewer's comment and have included this additional mouse model. We do not see a difference in COPD endpoints with this model. While there could be a host of molecular mechanisms behind these findings, it does indicate that E-cadherin is not involved in the function of these club cells.

E-cad deletion in different cell types resulted in different dysfunction in vivo. This is still the entrance of the study themes and can be enriched with more effort. I hope authors accumulate the more datasets and hope resubmit it to the journal.

We appreciate the reviewer's encouragement and hope that the addition of two new mouse models, one demonstrating protection, and evidence of cell-specific mechanisms of dysfunction, as well as additional data demonstrating altered cellular proliferation and differentiation in the normal and COPD cells, will sufficiently add to this already rather overwhelming manuscript.

Reviewer 3 Comments for the Author:

In testing the idea that E-cadherin reduction is causal to the emphysematous injuries, one of important experiments is to study whether overexpression of E-cadherin in COPD cells is sufficient to restore the normal epithelial structure and functions in these cells. To do this, authors used CS-exposed mTEC and CHEBEs, demonstrating significant recovery of their barrier function due to E-cadherin overexpression. However, they have never performed histological or cytological observations to look at what kinds of structural changes were induced in these cells. I suggest co-immunostaining of the cells for E-cadherin and a tight junction protein such as ZO-1, to confirm whether normal cell junctions, which had been lost in COPD cells, were morphologically restored by E-cadherin overexpression.

To provide a complete landscape of the various tight junction markers we have included the analysis of the transcriptional changes of Claudins (CLDN 1, 3, 7, 8, & 10), Occludin (OCLN), and tight junction proteins (TJP 1 & 2) in normal, COPD, and COPD with *CDH1* overexpression (Fig. 7D-K).

In some immunofluorescence images (e.g., Fig. 1 (1 month) and Fig. 3), it is very difficult to see whether E-cadherin level really differs between the control and experimental conditions. This problem should be addressed by providing better images or by adding a larger magnification to each of the image panels. To the Fig. 3B panels, the label 'BrdU' should be added.

We apologize for the difficulty seeing the differences and have included single-stained black and white images as well as the pseudo-colored merged images for improved visualization. In addition, we have included quantified intensities of the data.

Sincerely,

Venkataramana Sidhaye, MD
Associate Professor, Pulmonary and Critical Care Medicine
Johns Hopkins University School of Medicine
615 N. Wolfe St. E7626
Baltimore, MD 21205

Reviewers' comments:

Reviewer #1 (Remarks to the Author):

1. The authors have evolved the manuscript substantially, both with textual modifications and new data, since the first review round. It continues to be interesting and overall the changes have helped but at the expense of being highly complex and less easy to follow. What is the evidence for lung fibrosis in the first more global section of Results?
2. There is still the issue of whether the E-cad down-regulation is a model of "COPD" per se...I would stick with considerable neutrality on this: there are certainly a whole variety of pathological changes that occur under different specific circumstances, some of which are reminiscent of aspects of different human chronic diseases, from pan-acinar emphysema to lung fibrosis to peri-bronchial fibrosis, though the latter seems specific to E-cad down-regulation experiment in AT1 cells for example (and for some unexplained reason including that E-cad expression levels did not actually fall, if I am getting that right?). It would help the reader if in the early part of discussion there was a "list" of what path changes occur with what cellular alteration(s), and perhaps what disease entity you suggest that they most look like (but with humility on this).
3. It would also help readers if there was a list given of all abbreviations.
4. I liked the appearance of more data on EMT, and the more mature tone to Discussion that has emerged (absent the over-simplistic statement in line 4).
5. I don't think that "The" is needed as first word in Abstract.
6. What does cellular "jammed status" mean?

Reviewer #2 (Remarks to the Author):

I am satisfied with the author's responses to my questions/issues raised in my initial review. T

Reviewer #3 (Remarks to the Author):

The authors of this revised manuscript have done additional experiments, faithfully responding to reviewers, and also improved data presentation. Through these revisions, the points raised by this reviewer have satisfactorily been addressed. This is an important manuscript reporting the potential role of E-cadherin down-regulation in COPD, and I now recommend its publication in Communications Biology.

Reviewer 1 Comments for the Author:

1. The authors have evolved the manuscript substantially, both with textual modifications and new data, since the first review round. It continues to be interesting and overall the changes have helped but at the expense of being highly complex and less easy to follow. What is the evidence for lung fibrosis in the first more global section of Results?

We appreciate that the reviewer finds the topic interesting and acknowledge that the manuscript has become more complex with the additional data included but hope this also addresses the biology. The reviewer is correct in that the evidence of lung fibrosis in the global model is minimal, and this should not be included in this section. We have deleted it.

2. There is still the issue of whether the E-cad down-regulation is a model of "COPD" per se...I would stick with considerable neutrality on this: there are certainly a whole variety of pathological changes that occur under different specific circumstances, some of which are reminiscent of aspects of different human chronic diseases, from pan-acinar emphysema to lung fibrosis to peri-bronchial fibrosis, though the latter seems specific to E-cad down-regulation experiment in AT1 cells for example (and for some unexplained reason including that E-cad expression levels did not actually fall, if I am getting that right?). It would help the reader if in the early part of discussion there was a "list" of what path changes occur with what cellular alteration(s), and perhaps

what disease entity you suggest that they most look like (but with humility on this).

We appreciate the details the reviewer raises and have tried to take a more neutral tone and discuss tissue disruption and remodeling instead of "COPD," which is a clinical diagnosis. In addition, we attempted to "list" the disease entity that the model most resembles by balancing the fact that it is not, in fact entirely identical to the disease.

3. It would also help readers if there was a list given of all abbreviations.
Following the Journal policies, we have defined all abbreviations at first use.
4. I liked the appearance of more data on EMT, and the more mature tone to Discussion that has emerged (absent the over-simplistic statement in line 4).
We appreciate that the reviewer is interested in the EMT data and have tried to modulate the over-simplistic statement.
5. I don't think that "The" is needed as first word in Abstract.
We have removed it.
6. What does cellular "jammed status" mean?
We apologize for not explaining this word clearly, the use of which is not appropriate for the metrics demonstrated in this manuscript. We have changed the language to cell velocity and movement, a better description of the data presented.

Sincerely,

Venkataramana Sidhaye, MD
Associate Professor, Pulmonary and Critical Care Medicine
Johns Hopkins University School of Medicine
615 N. Wolfe St. E7626
Baltimore, MD 21205